# An implantable soft robotic ventilator augments inspiration in a pig model of respiratory insufficiency

**Lucy Hu**[1,2], **Jean Bonnemain** [2,3], **Mossab Y. Saeed** [4], **Manisha Singh**[2], **Diego Quevedo Moreno** [5], **Nikolay V. Vasilyev**[4] & **Ellen T. Roche** [2,5] ✉

Severe diaphragm dysfunction can lead to respiratory failure and to the need for permanent mechanical ventilation. Yet permanent tethering to a mechanical ventilator through the mouth or via tracheostomy can hinder a patient's speech, swallowing ability and mobility. Here we show, in a porcine model of varied respiratory insufficiency, that a contractile soft robotic actuator implanted above the diaphragm augments its motion during inspiration. Synchronized actuation of the diaphragm-assist implant with the native respiratory effort increased tidal volumes and maintained ventilation flow rates within the normal range. Robotic implants that intervene at the diaphragm rather than at the upper airway and that augment physiological metrics of ventilation may restore respiratory performance without sacrificing quality of life.

The diaphragm is the major muscle responsible for inspiration and contributes up to 70% of the inspiratory tidal volume in a healthy individual[1,2]. Diaphragm dysfunction can result from a variety of etiologies including phrenic nerve trauma[3] and neuromuscular disease[4,5]. Owing to the degenerative nature of many of these etiologies, mechanical respiratory failure exists as a continuous spectrum of dysfunction. Severe diaphragm dysfunction or paralysis can lead to chronic respiratory failure. When disease progresses beyond the treatment capacity of non-invasive treatment, patients must make the difficult decision to opt for permanent invasive ventilation via a tracheostomy or to pursue palliative care with an understanding of the terminal nature of their disease. Invasive ventilation can interfere with many aspects of a patient's quality of life, such as hindering speech, requiring full-time care and possibly necessitating the patient move into a care facility. There is an urgent need for therapeutic ventilation options that restore respiratory performance without sacrificing quality of life, especially for those with the most severe cases of diaphragm dysfunction.

Respiration is a fundamentally mechanical process. The diaphragm is a dome-shaped muscle that drives up to 70% of respiration[1,6].

Soft robotic actuators are ideal for reproducing complex, repetitive muscle contractions, such as that of the diaphragm, while interfacing non-destructively with biological tissue. Previously, fully implanted soft actuators have shown the ability to augment heart function[7–11] and many other newly developed implantable robotics have shown utility in a broad spread of biological applications[12–20]. Due to the mechanical nature of respiratory failure, especially in the context of conditions such as muscular dystrophy, implanted soft robotic actuators applied to the diaphragm have the potential to mechanically support and augment its function. There is minimal previous work investigating soft robotics applied to the augmentation of respiration; one of the few examples reports a dielectric elastomer sheet used to completely replace an excised diaphragm and generate motion[12,21]. Contrastingly, the work presented here leaves the native diaphragm intact while demonstrating function in terms of augmentation of clinically relevant physiological metrics (ventilation flows, volumes and pressures) in addition to diaphragm motion in an in vivo porcine model as opposed to solely replicating diaphragm motion while excising the native diaphragm.

[1]Harvard-MIT Program in Health Sciences and Technology, Massachusetts Institute of Technology, Cambridge, MA, USA. [2]Institute for Medical Engineering and Science, Massachusetts Institute of Technology, Cambridge, MA, USA. [3]Department of Adult Intensive Care Medicine, Lausanne University Hospital and University of Lausanne, Lausanne, Switzerland. [4]Department of Cardiac Surgery, Boston Children's Hospital, Harvard Medical School, Boston, MA, USA. [5]Department of Mechanical Engineering, Massachusetts Institute of Technology, Cambridge, MA, USA. ✉e-mail: etr@mit.edu

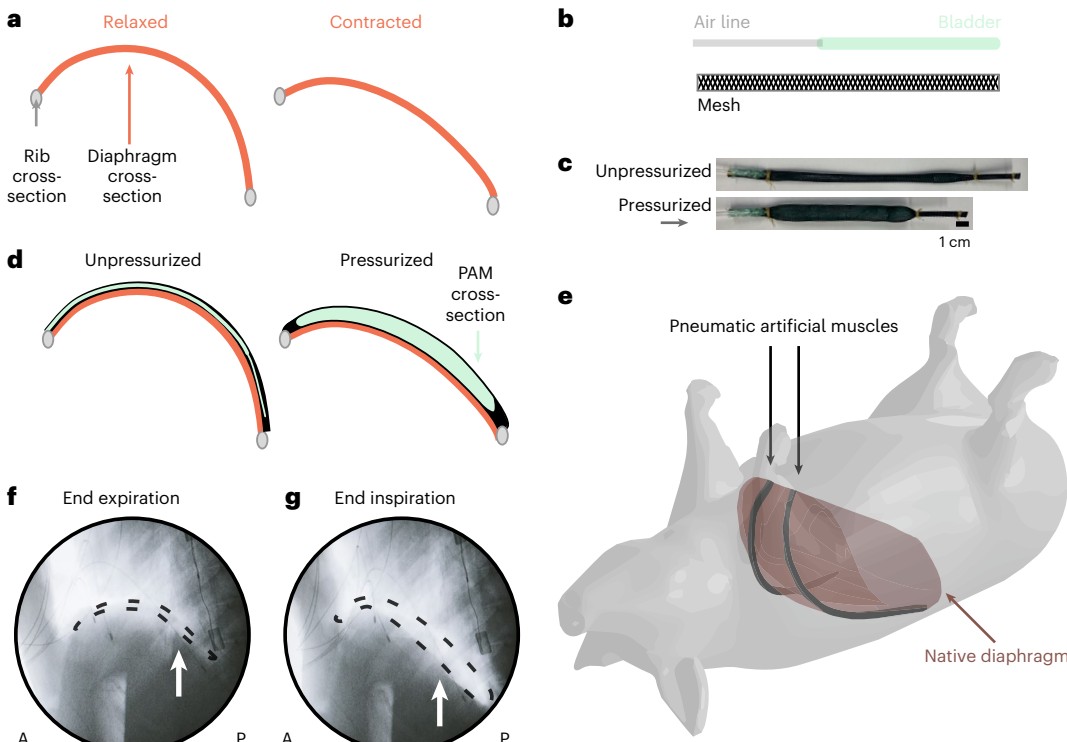

**Fig. 1 | Overview of the use of implantable PAMs for augmenting respiratory muscle function. a**, Schematic depicting the lateral cross-section of the native diaphragm anchored to the ribs in a relaxed (left) and contracted (right) state. **b**, Schematic of the components that makeup a single PAM. **c**, Pictures of a single PAM in an unpressurized and pressurized state. **d**, Lateral cross-sectional schematic of the strategy to augment diaphragm motion by placing PAMs superior to the diaphragm. The PAM conforms to the relaxed diaphragm in its unpressurized (left) state and pushes the diaphragm caudally in its pressurized (right) state. **e**, Visualization of the placement of PAMs (in black) superior to the diaphragm in a live pig model. **f,g**, Lateral fluoroscopy view of the in vivo porcine diaphragm with PAMs in an unpressurized (**f**) and pressurized (**g**) state (fluoroscopic videos available as Supplementary Video 1). The air-filled balloon of the actuator is outlined with a dashed line and indicated by an arrow. *A* and *P* denote the anterior and posterior direction of the animal, respectively.

Here we demonstrate a diaphragm-assist system that functions as an implantable ventilator by using soft robotic actuators to mechanically augment diaphragm function during inhalation, increasing inspiration. As a proof-of-concept, we simulate a range of respiratory insufficiency within each animal—specifically, we induce respiratory depression via anaesthetics and diaphragm paralysis by severing the phrenic nerve—and then demonstrate the ability of the assist system to augment respiratory flows, volumes and pressures. We also investigate specific metrics of inspiratory function, including peak inspiratory flow and transdiaphragmatic pressure[22]. We show that to achieve effective inspiration assistance, the actuation of the assist system must be synchronized to the subject's underlying respiratory effort. To achieve this, we have built a control system in which actuation is triggered by the beginning of inspiration. Through an analysis of the respiratory waveforms, we investigate the optimal alignment of actuation with the subject's native respiratory effort. By augmenting diaphragm function in a biomimetic fashion, we demonstrate the replication and augmentation of the native biomechanics of respiration in which a negative pleural and alveolar pressure drives airflow, as opposed to the positive pressure ventilation of standard mechanical ventilation.

## Results

### Soft robotic design strategy applied to mechanically assisting inspiration

As depicted in the schematic in Fig. 1a, when the diaphragm contracts, the arclength of the diaphragm shortens, and the entire sheet of the diaphragm moves downwards, acting as a pump. The thoracic cavity volume increases and pressure decreases, ultimately driving respiration.

Our strategy aims to harness the contractile function of pneumatic artificial muscles (PAMs) to mimic and augment the native contraction of the diaphragm. We opt for a McKibben type PAM—a classical soft actuator type with a simple fabrication process and high force generation[23,24] that is capable of mimicking and augmenting biological systems[7,8,13]. At their simplest, McKibben actuators are composed of an expandable weaved mesh surrounding a bladder connected to an air line (Fig. 1b) (Methods). When the bladder is pressurized, the mesh expands radially and drives linear contraction (Fig. 1c). The McKibben actuators used in this work can generate up to 40 N of contractile force under 20 psi pressurization (Extended Data Figs. 1 and 2 and Supplementary Notes). Conceptually, we harness the linear contraction of these PAMs by placing them superior to the native diaphragm so that the relaxed PAM conforms to the native curvature of the diaphragm (Fig. 1d). Mimicking the native diaphragm, we anchor the ends of the PAMs to the ribs (Methods). With pressurization, the length of the PAM shortens, the arclength shortens and the PAM mechanically pushes the diaphragm downwards (shown in situ in Supplementary Fig. 1). Actuator behaviour is governed by the degree of pressurization. Set pressurization waveforms are programmed to the control system and electropneumatic regulators. In vitro and in vivo characterization of actuator behaviour when controlled by different pressurization waveforms is included in Extended Data Figs. 1 and 2.

In contrast to the dielectric artificial diaphragm[12], our diaphragm-assist system uses a set of two linear PAMs, leaves the native diaphragm intact and has a low-profile presence (deflated: 5 ml volume, inflated: 17 ml volume). To test this concept in a live porcine model, we surgically implanted a pair of McKibben actuators in

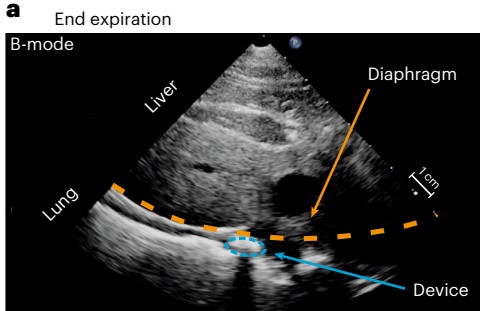
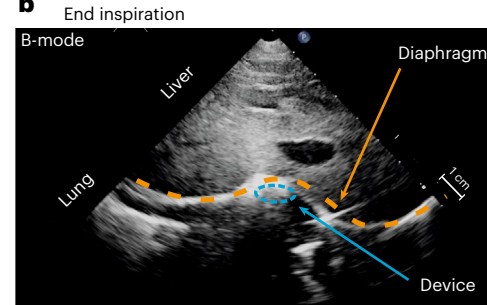

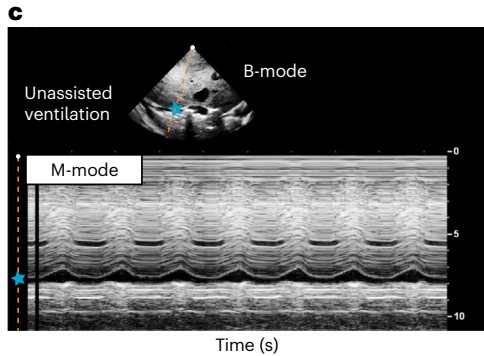
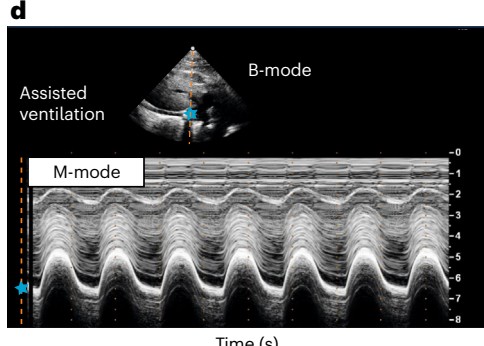

**Fig. 2 | Ultrasound imaging of the diaphragm and its associated displacement with and without assisted ventilation. a,b**, Two-dimensional view (B-mode) of the diaphragm at the end of expiration (**a**) (device not pressurized, muscle relaxed) and at the end of inspiration (**b**) (device pressurized, muscle contracted). **c,d**, M-mode evaluation of diaphragm motion during unassisted ventilation (**c**) and assisted ventilation (**d**) (20 psi). For all images, the probe was positioned in the right subcostal space, pointing toward the cranial direction. Orange dashed line, diaphragm; blue dashed ellipse, actuator cross-section; blue star, spatial position of the diaphragm.

an anterior-to-posterior direction lateral to the heart. The actuator placement is visualized in a three-dimensional rendering in Fig. 1e. Fluoroscopy of the diaphragm was taken throughout the experiments. The lateral cross-sectional view from the fluoroscopy shows the realization of our soft robotic strategy in an in vivo pig model (Fig. 1f,g).

The actuators push the diaphragm caudally, augmenting the diaphragm displacement. Ultrasonography is used to visualize and quantify diaphragm displacement (Fig. 2). The coronal plane cross-section of the actuator and the diaphragm is visualized via two-dimensional (B-mode, brightness) ultrasonography of the diaphragm (Fig. 2a,b). To quantify the motion of the device and the diaphragm, we used M-mode (motion) ultrasonography (Fig. 2c,d), which visualizes the image along a single line, selected within the B-mode image, over time. The M-mode features excellent axial and temporal resolutions and is particularly well-suited for motion analysis[25]. The actuator augments the diaphragm displacement per breath from 0.37 cm displacement of unassisted ventilation (Fig. 2c) to 1.92 cm displacement of assisted ventilation (Fig. 2d).

**Augmenting tidal volume and peak inspiratory flow in vivo**
To evaluate the ability of our diaphragm-assist system to augment respiratory function, the animals were instrumented to collect physiological data, including respiratory flows, volumes and pressures within the respiratory system (Supplementary Fig. 2). The pressurization of the soft robotic actuators was controlled via a custom-built control system; the actuation pressure data were input into the same high-resolution data acquisition system as the physiological data (Methods).

Ventilation is key to driving $CO_2$ exchange, so we first examine the flow and volume waveforms as metrics of ventilatory function. Flow is measured by a spirometer. Peak inspiratory flow can be used as a clinical metric of inspiratory function[22], which yields a direct measurement of the effect of the diaphragm-assist system. Integrating the flow with respect to time yields a volume waveform over time. The volume of

each breath (tidal volume) and its rate (minute ventilation) are the most relevant parameters in directly measuring ventilation. Pressures within the respiratory system, such as pleural and abdominal pressures, reveal information about the respiratory biomechanics that physically drive ventilation and are discussed later in this work.

To start each study, the animal was anaesthetized appropriately with isoflurane and placed on mechanical ventilation. Isoflurane induces a respiratory depression with decreased tidal volumes and increased respiratory rate that ultimately combine to a reduced minute ventilation[26]. The respiratory depression secondary to the isoflurane is used as our baseline animal model of respiratory insufficiency due to hypoventilation. Each subject has a reduced but non-zero respiratory drive and response to $CO_2$. Mechanical ventilation is used to support the animal throughout the implantation surgery. Within each subject, we introduce a series of respiratory challenges, collecting data during periods of unassisted ventilation (in which any spontaneous respiration is due to the native respiratory drive) and during periods of actuator-assisted ventilation. Mechanical ventilation is used to restore and maintain a state of normoventilation after and between respiratory challenges. To investigate the effect of the diaphragm-assist system, a representative respiratory challenge was chosen per subject. The phrenic nerve is intact for all data shown in Fig. 3.

In a vignette from the best-responding subject (Fig. 3a), we show that the assist system has the direct capacity to augment the peak inspiratory flow from 0.18 l s$^{-1}$ to 0.59 l s$^{-1}$ and the tidal volume from 55 ml to 161 ml. When the assist is resumed after a short period of unassisted respiration, the augmentation effect of the actuation on the flow and volume waveforms is re-established nearly immediately over the course of 2 breaths.

An example of a full respiratory challenge is shown in Fig. 3b. During the unassisted ventilation at the start of the challenge, the subject models a state of hypoventilation. During this period, the tidal volumes and flows have a slight increase over time, indicating

that the baseline respiratory drive is responding to the increasing $CO_2$ status due to the unassisted low minute ventilation (0.9 l min$^{-1}$). When assist is switched on (as indicated by the actuator pressure waveform, the white background and the black arrow), there is a clear jump in the peak inspiratory flow (+0.20 l s$^{-1}$, 95% CI: +0.19 l s$^{-1}$ to +0.22 l s$^{-1}$), tidal volumes (63 ml, 95% CI: 58 ml to 68 ml) and minute ventilation (0.9 l min$^{-1}$ to 3.1 l min$^{-1}$). The actuators cycle between a pressurized and unpressurized state for 10 min. At the end of the respiratory challenge when the respiratory effort has reached a steady state, the assist is switched off and we see that the respiratory effort drops slightly (peak inspiratory flow: −0.09 l s$^{-1}$, 95% CI: −0.08 to −0.10; tidal volume: −10 ml, 95% CI: −7 to −13 ml) but much less than the jump seen at the start of the respiratory challenge.

The respiratory drive is a slow but dynamic factor underlying all of the respiratory physiology data. As seen in the first 200 s of Fig. 3b, the respiratory drive visibly increases as the low minute ventilation leads to $CO_2$ buildup. This response to $CO_2$ is dynamic and varies between subjects on the basis of each animal's response to isoflurane. By examining the breaths immediately before and after these transition points (off-to-on and on-to-off), we can examine the direct effect of the diaphragm-assist system in terms of augmenting volume and peak inspiratory flow while minimizing the influence of the changing baseline.

This analysis was conducted for one representative respiratory challenge for each of 5 subjects. These 5 subjects represent the subset of experiments conducted with uninterrupted respiratory challenges that collected the unassisted baseline data both before and after device assistance. We see a spectrum of responsiveness to the diaphragm-assist system across subjects (Fig. 3c–e). The subjects are ordered from largest change in tidal volume at the start of the challenge to the smallest (best responder to worst responder according to Fig. 3d). We find that the diaphragm-assist system generates much larger respiratory augmentations at the beginning of a trial—when mechanical ventilation support has just been removed, minute ventilation drops suddenly and the animal's $CO_2$ state rises rapidly—than at the end of the respiratory challenge when the respiratory baseline is relatively more stabilized (Fig. 3c–e).

Subject A was much more responsive to the assist system than any other subject. In terms of tidal volume, 4 of the 5 subjects show an augmentation of >30 ml per breath at the beginning, whereas only 1 of the subjects shows substantial augmentation to the tidal volume at the end. Of the 4 less responsive subjects (B,C,D,E), 3 show a mild response at the end while in the worst responder (E), the actuation overall decreased the ventilation metrics (Fig. 3c–e). The subject with

the weakest response had the highest baseline weight-normalized minute ventilation at the beginning of the trial (Fig. 3e) compared with other subjects.

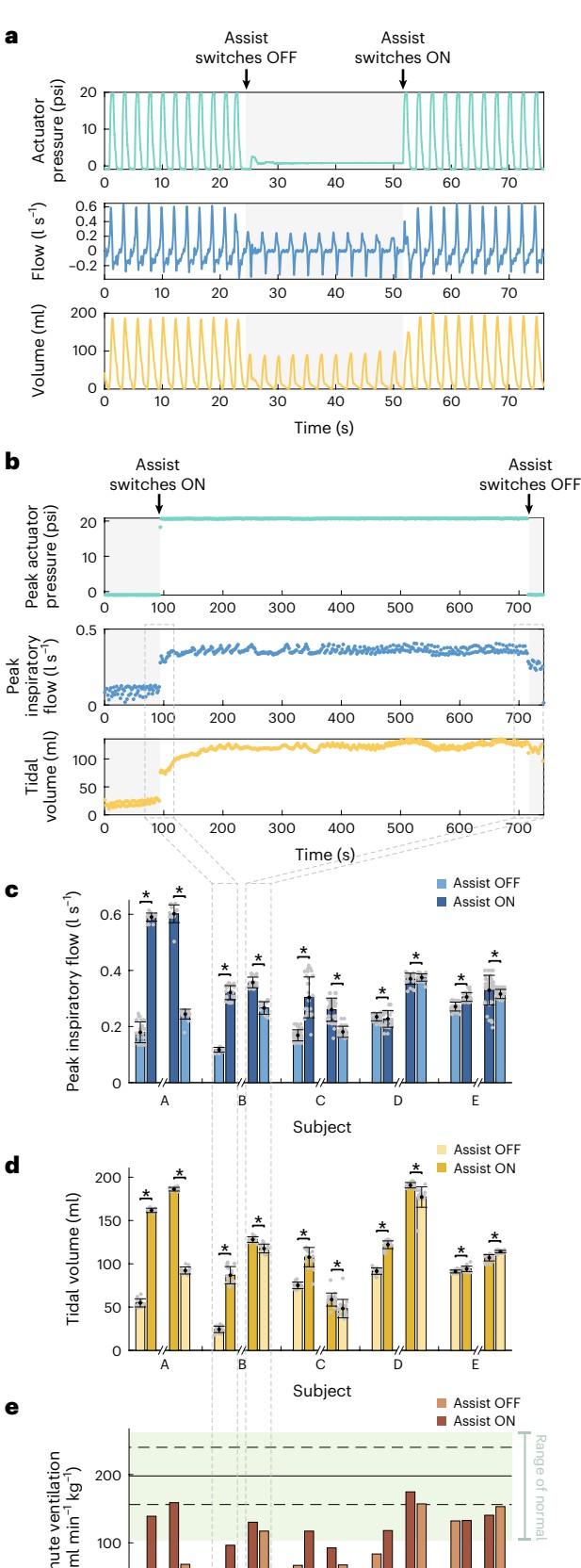

**Fig. 3 | Ability to augment tidal volume and peak inspiratory flow in vivo.**
**a**, A representative continuous segment of actuation pressure, flow and tidal volume waveforms from the respiratory challenge with the largest augmentation. Grey shading indicates the period when the diaphragm-assist system is off, and the subject's respiration is unsupported. **b**, A representative set of peak actuation pressure, peak inspiratory flow and tidal volumes for one full respiratory challenge. Grey shading indicates the period when the system is off, and respiration is unassisted. **a** and **b** represent 1 biological replicate. **c,d**, Comparison of the average peak inspiratory flow (**c**) and tidal volume (**d**) in the 30 s period immediately before and after the point where the assist is turned on at the beginning (left two bars per subject) and off at the end (right two bars per subject) of the respiratory challenge (as represented by the arrows in **b** and the grey dashed lines in **b–e**) across 5 independent biological replicates (subjects A–E, with 11–27 breaths per subject). Each grey dot on the plots represents technical replicates (1 breath) within the subjects. **e**, Body weight-normalized minute ventilation achieved during the 30 s period immediately before and after the assist is turned on at the beginning and off at the end of the respiratory challenge across 5 independent biological replicates (subjects A–E). The range of normal minute ventilation, as reported in ref. [27], is indicated by the light green shading; the solid and dashed lines indicate the mean ± s.d. In **c** and **d**, bar plots and error bars show the mean ± s.d., *$P < 0.001$ using a two-sided Wilcoxon rank-sum test.

Body weight-normalized minute ventilation is used to compare these results to normal physiology. Minute ventilation is a metric of the ventilation rate, taking into account both tidal volume and the respiratory rate. In a normal, conscious pig, the expected body weight-normalized minute ventilation is 198 ml min⁻¹ kg⁻¹ ± 41 ml min⁻¹ kg⁻¹ with a range of 104 ml min⁻¹ kg⁻¹ to 262 ml min⁻¹ kg⁻¹ (ref. [27]), indicated by the green shading in Fig. 3e. Actuator-assisted ventilation allowed all 5 subjects to reach the lower range of normal physiology, and 2 of the subjects even achieved a minute ventilation corresponding to one standard deviation below the normal mean (Fig. 3e). However, we note that this minute ventilation is achieved with low tidal volumes and high respiratory rates, which results in a lower alveolar ventilation than the same minute ventilation achieved with high tidal volumes and low respiratory rates.

## Synchronizing with the underlying respiratory effort

As with standard mechanical ventilation[28,29], patient–ventilator synchrony in our system is critical to the ability to augment respiration. Asynchronous ventilation can destructively interfere with the underlying respiratory effort, leading to worse ventilation with assistance than without.

To synchronize the actuation of our assist system with the subject's underlying respiratory effort, we built a control system (Fig. 4a,b) that can actuate on the basis of the respiratory flow rate. The system uses the spirometry flow sensor as the source data. The flow data are read into our data acquisition system. The associated data analysis software allows a user-set threshold voltage; this threshold voltage is manually titrated during every respiratory trial to achieve qualitatively good synchronization. When the flow rate passes this set threshold, a digital pulse is triggered and sent to the microcontroller in our control box. The microcontroller triggers a pre-set actuation pressure waveform of one cycle of pressurization and depressurization in the electropneumatic regulator, filling and emptying the PAMs with pressurized air (further details in Methods).

Our control system can implement both a set, rhythmic control scheme independent of the native respiratory effort or a dynamic control scheme synchronized with the underlying respiratory effort. Due to the phase and frequency mismatch between the independent actuation and the underlying respiratory effort, the mixed interference of the actuator and the underlying respiratory effort can be seen in both the flow and volume waveform (Fig. 4c). Contrastingly, the well-synchronized actuation reveals much more homogeneous flow and volume waveforms (Fig. 4d).

Within each subject, we compare the tidal volumes and peak inspiratory flows from one representative challenge of independent actuation with one representative challenge of synchronized actuation (details in Methods). We find that synchronized actuation consistently produces much less variance in the tidal volumes (Fig. 4e,f). Although in some subjects, such as subject A, independent actuation achieved a few higher maximum tidal volumes, the independent actuation also achieved lower minimum tidal volumes across all subjects due to the misalignment of actuations with the underlying respiratory effort leading to destructive interference, or due to actuation with no underlying breath, representing a breath that is solely actuator driven. Misalignment between the diaphragmatic contraction and the device during independent actuation can be observed with M-mode ultrasound (Fig. 4g), in contrast to synchronized actuation (Fig. 4h). Asynchronous moments of native diaphragm contraction produce a heterogeneous waveform, as indicated by the orange arrows in Fig. 4g.

## Effect of synchronization on blood gas exchange

Physiologically, ventilation is necessary to bring in oxygen ($O_2$) and to clear out accumulated carbon dioxide ($CO_2$) from the blood. Arterial blood gases (ABGs) are discrete blood analyses that give a snapshot view of the gas exchange and acid-base homoeostasis, providing measurement of partial pressures of $O_2$ ($P_aO_2$) and CO2 ($P_aCO_2$), pH and bicarbonates ($HCO_3^-$) in arterial blood. $P_aCO_2$ is directly and inversely proportional to alveolar ventilation and is therefore a representative metric of ventilatory function. Only pH and $p_{CO_2}$ are depicted here in Fig. 5, but the full ABG parameters are reported in Supplementary Table 1 and discussed in Supplementary Notes.

As shown in the previous section, the high variance from independently actuated ventilation showed mixed constructive and destructive interference (Fig. 4e,f) which led to worse ventilation outcomes. The same variance in the peak inspiratory flows and tidal volumes over time due to independent vs synchronized actuation can be seen in Fig. 5a,b. In these two respiratory challenges, a single subject was switched directly from the standard mechanical ventilation to our diaphragm-assist system, evaluating its ability to maintain gas exchange.

In the respiratory challenge operated with independent actuation (Fig. 5a), we see high levels of hypercarbia over time. As a result, respiratory acidosis develops, which is a direct consequence of increased $P_aCO_2$ (Supplementary Table 1a). Contrastingly, in a respiratory challenge operated with synchronized actuation in the same animal (Fig. 5b), $p_{CO_2}$ levels are relatively well maintained. The acidemia observed for this trial is a combination of respiratory and metabolic causes (called mixed acidosis), with a predominant respiratory component (Supplementary Table 1b and Supplementary Notes).

In another experiment on a different subject, a respiratory trial was initiated with 2 min of unsupported ventilation and then switched to our diaphragm-assist system, evaluating its ability to recover from a period of unsupported ventilation. During the 2 min of unsupported ventilation, high levels of $CO_2$ accumulate quickly over this brief amount of time (Fig. 5c). After 2 min, the diaphragm-assist system is actuated with synchronized actuation. The increasing acidification and accumulation of $CO_2$ reverses and some recovery from the hypercarbic state is seen in the first 10 min, with a slight uptick in $CO_2$ around 15 min into the challenge.

## Factors in optimizing synchronization

As seen by the mixed interference in Fig. 4c,g and the ability of independent actuation to maintain blood gas balance in Fig. 5a, the alignment of the actuation with the underlying respiratory effort will critically determine the constructive versus destructive nature of the interference. In respiratory challenges that had an independent actuation scheme or a poorly synchronized actuation scheme, we found the datasets that provide a natural variation in the timing of the actuation in relationship to the underlying respiratory effort.

Because mechanical respiratory failure exists as a continuous spectrum of loss of function, we looked at the implications of synchronization at different levels of baseline respiratory effort. As seen in Fig. 3, there is variance in the underlying respiratory function between subjects. To simulate a controlled change in the underlying respiratory function within the same subject, we severed the phrenic nerve in some subjects, simulating diaphragm paralysis, in combination with the respiratory depression due to isoflurane (Methods). Fig. 6 depicts the analysis of aligning the actuator synchronization to the underlying respiratory effort for two respiratory challenges within subject B: (1) the subject with preserved diaphragm function (Fig. 6, left) and (2) the subject with a severed phrenic nerve (Fig. 6, right).

To optimize for maximum inspiratory augmentation, we investigate the relationship of the timing of different waveform features to the resulting tidal volume and peak inspiratory flow of each breath. The high frequency sampling of our data acquisition system (1,000 Hz) allows for millisecond temporal resolution. Custom software was written to analyse the actuation pressure, flow and volume data.

We identify the breath bounds as determined by the local minima in the volume waveform (the locations of $V_0$), and then find the time distance between identified waveform features for each individual breath (further details in Methods). Waveform features analysed include the start of an actuation waveform ($P_0$), peak inspiratory flow ($F_{pk}$), the start

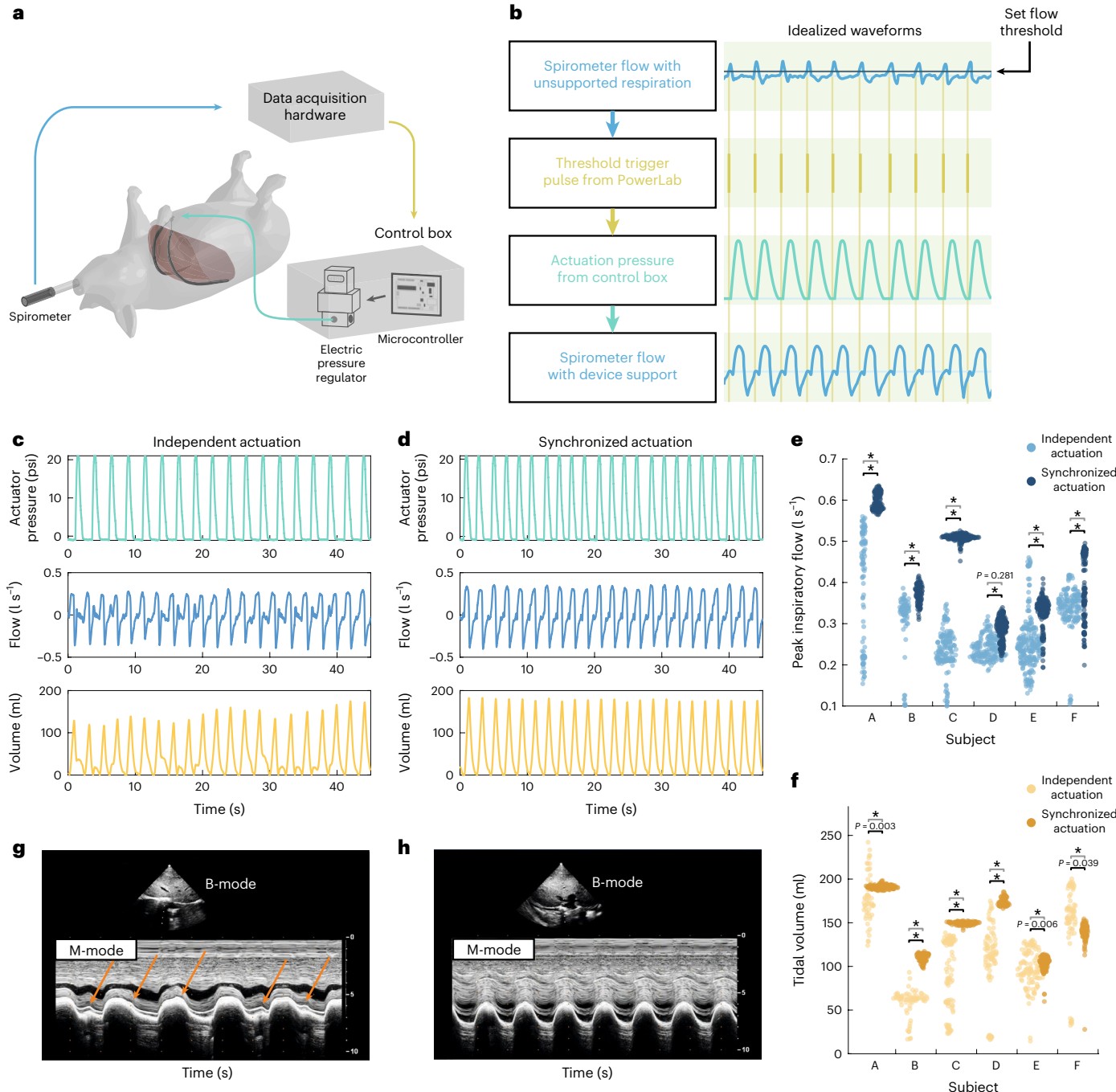

**Fig. 4 | Synchronous actuation with the native respiratory effort. a**, Schematic of the control system. The spirometry flow sensor data are fed into the data acquisition system; when the flow sensor crosses a set threshold, a trigger pulse is sent to the control box which triggers a set pressure actuation curve in the electropneumatic regulator, modulating the pressure inside the PAMs. **b**, A set of idealized waveforms (indicated by the green background) showing the mechanism of synchronization. **c,d**, A representative set of collected waveform data (actuation pressure, flow and tidal volume) from 1 subject for a set independent actuation scheme (**c**) and a synchronized actuation scheme (**d**). **e,f**, A swarm plot comparing the steady-state tidal volumes (**f**) and peak inspiratory flows (**e**) generated with independent actuation (light blue and light yellow) and with synchronized actuation (dark blue and orange) for 6 independent biological replicates (subjects A–F, with 119–419 breaths per subject). Each dot on the plots represents technical replicates (1 breath) within the subjects. **g,h**, M-mode analysis during independent (**g**) or synchronized (**h**) actuation. Orange arrows point toward asynchronous diaphragmatic muscle contraction. In **e** and **f**, steady-state data are taken from 5 min to the end of the respiratory challenge. Black significance bars are results from two-sided Welch's *t*-test comparing means. Grey significance bars are results from a two-sample *F*-test for equal variances, *$P < 0.001$ for both statistical tests.

of inspiration ($V_0$), the start of expiration ($V_{pk}$) and others (Fig. 6a,b and Supplementary Fig. 3).

The distances between features act as different metrics of alignment and elucidate what factors are important to consider in optimizing synchronization. There are many different features and feature distances that can be analysed. Fig. 6c–f shows the time relationship of the start of expiration to the actuation pressure ($V_{pk}$-$P_0$), but other metrics are shown in Supplementary Fig. 3.

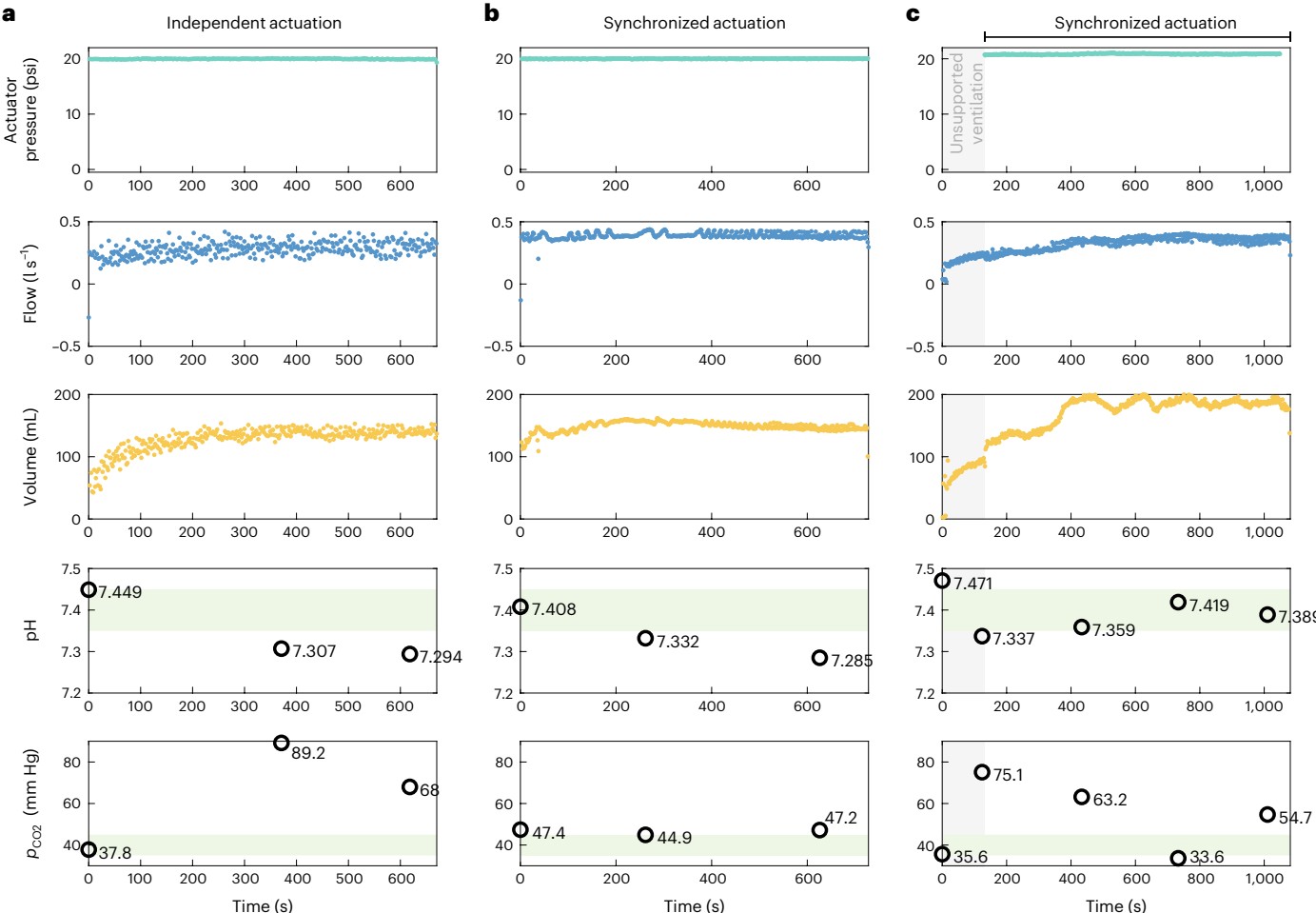

**Fig. 5 | ABGs taken across distinct respiratory challenges. a**, In a respiratory challenge operated with independent actuation, a representative set of peak actuation pressures, peak inspiratory flows and tidal volumes, and the pH and $p_{CO2}$ values from discrete arterial blood gases are shown. **b**, In a respiratory challenge operated with synchronized actuation, a representative set of peak actuation pressures, peak inspiratory flows and tidal volumes, and the pH and $p_{CO2}$ values from discrete arterial blood gases taken during one full respiratory challenge are shown. The respiratory challenges depicted in **a** and **b** are taken from the same subject (1 biological replicate). **c**, In another animal (1 biological replicate), a respiratory challenge began with a 2 min period of unsupported ventilation and subsequent synchronized actuation. A representative set of peak actuation pressures, peak inspiratory flows and tidal volumes, and the pH and $p_{CO2}$ values from discrete arterial blood gases are shown. Grey shading in **c** indicates the period when the system is off and respiration is unassisted. In the bottom rows of **a**–**c**, light green shading indicates the standard range of normal values for each arterial blood gas metric. Complete ABGs can be found in Supplementary Table 1.

We examine the influence of these time metrics on tidal volume and peak inspiratory flow. We find that the most important predictor variables are time metrics related to the start of expiration ($V_{pk}$). With diaphragm function preserved, there is a weak linear relationship between $V_{pk}$-$P_0$ and the peak inspiratory flow ($R^2 = 0.31, P < 0.001$) (Fig. 6c), and no correlation with the tidal volume ($R^2 = 0.04, P = 0.001$) (Fig. 6e). However, when the diaphragm function is removed by severing the phrenic nerve, a clear linear relationship emerges between $V_{pk}$-$P_0$ and tidal volume ($R^2 = 0.84, P < 0.001$) (Fig. 6f) and a weaker relationship with peak inspiratory flow ($R^2 = 0.30, P < 0.001$) (Fig. 6d).

Notably, we do not find these relationships when using the timing between the start of actuation and the start of inspiration ($P_0$-$V_0$) as a metric. There is no linear relationship between $P_0$-$V_0$ and the peak inspiratory flow or tidal volume for both the cases with and without diaphragm function (Supplementary Fig. 4).

## Comparing respiratory biomechanics

To compare the respiratory biomechanics of different modes of respiration and ventilation, pleural pressure ($P_{pl}$), abdominal pressure ($P_{ab}$) and transdiaphragmatic pressure ($P_{di}$; $P_{di} = P_{ab} - P_{pl}$) waveforms are analysed. Transdiaphragmatic pressure is a metric of diaphragm function[6,30,31]. Pleural pressure and abdominal pressure are approximated by a sensor mounted on a balloon catheter placed in the oesophagus and stomach, respectively. As these sensors approximate $P_{pl}$ and $P_{ab}$, the measurements are interpreted as relative measurements and not absolute ones (see Methods for information about instrumentation and normalization). When analysing relative pressure waveforms, the most informative metric is the maximum change in pressure per breath.

In Fig. 7a–c, we show that across subjects (subject C was not instrumented for pressure measurements, and is therefore not shown), actuator-assisted ventilation more closely matches the respiratory biomechanics of spontaneous respiration than mechanical ventilation. Mechanical ventilation pushes air into the lungs, increasing pleural pressure with inspiration, whereas both actuator-assisted ventilation and spontaneous respiration generate a negative pleural pressure to drive airflow. As the diaphragm is passive in mechanical ventilation, we see a negligible change in the abdominal pressure, whereas the caudal

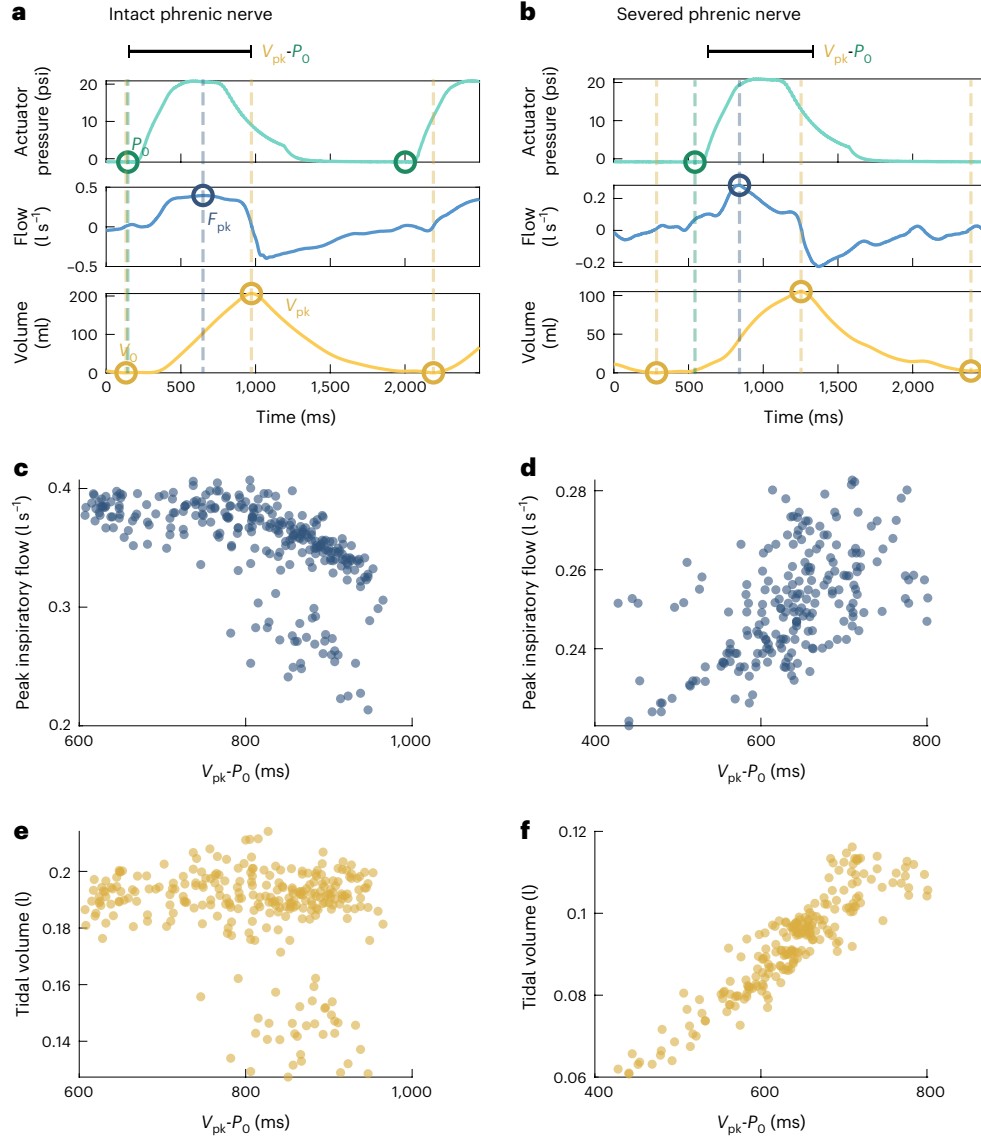

**Fig. 6 | Effect of the time alignment of actuation and native respiratory effort in two levels of respiratory insufficiency. a,b**, Representative actuation pressure, flow and volume waveforms for a single breath from one respiratory challenge for a single subject (1 biological replicate) with an intact phrenic nerve (**a**) and one challenge with a severed phrenic nerve (**b**). Circles mark features that can be identified from the waveforms including the start of actuation ($P_0$), peak inspiratory flow ($F_{pk}$), start of the breath ($V_0$) and peak volume ($V_{pk}$), with the dashed lines indicating the time point of each feature. **c,d**, A scatterplot of peak

inspiratory volume as it relates to the time between $V_{pk}$ and $P_0$ for one respiratory challenge with an intact phrenic nerve (**c**) (278 breaths) and one challenge with a severed phrenic nerve (**d**) (215 breaths). **e,f**, A scatterplot of tidal volume as it relates to the time between $V_{pk}$ and $P_0$ for one respiratory challenge with an intact phrenic nerve (**e**) (278 breaths) and one challenge with a severed phrenic nerve (**f**) (215 breaths). All data are taken from the same subject (1 biological replicate). Each dot represents 1 technical replicate (1 breath).

movement of the diaphragm in both actuator-assisted ventilation and spontaneous respiration increases abdominal pressure.

In the representative waveforms from subject A (Fig. 7d–f)— the case of highest responsiveness as seen in Fig. 3c–e—the actuator-assisted ventilation not only more closely resembles that of spontaneous respiration, but also augments all of the pressure waveforms. Actuator-assisted ventilation generates more negative changes in pleural pressure, greater increases in abdominal pressure and ultimately greater increases in transdiaphragmatic pressure per breath.

A graphical technique used to measure work of breathing (WOB) is the Campbell diagram, referencing pleural pressure with lung volume. Using the pressure and volume data from subject A, we generate the pressure-volume (PV) loops of a Campbell diagram (Fig. 7g). WOB is calculated from this PV loop as the internal area between the inspiratory edge of the loop and the passive chest wall compliance derived from the

mechanical ventilation PV data. Normal WOB is 0.35–0.7 J l⁻¹ (refs. [22,32,33]). During attenuated spontaneous breathing, the subject's WOB is 0.10 J l⁻¹. During actuator-assisted ventilation, the assist system shares the WOB and increases the total average WOB to 0.17 J l⁻¹, a 66% increase.

## Discussion

We have used pneumatic soft robotic actuators to support and augment respiration, demonstrating acute augmentation of physiological metrics of respiration and feasibility as a proof-of-concept device. A set of two McKibben-style PAMs surgically implanted superior to the diaphragm can provide mechanical support to the diaphragm in a large animal model of respiratory insufficiency. We thoroughly characterized the in vitro mechanical properties of the device and investigated its interactions with the respiratory system and the subject, using multimodal metrics to evaluate respiratory function (in particular, tidal

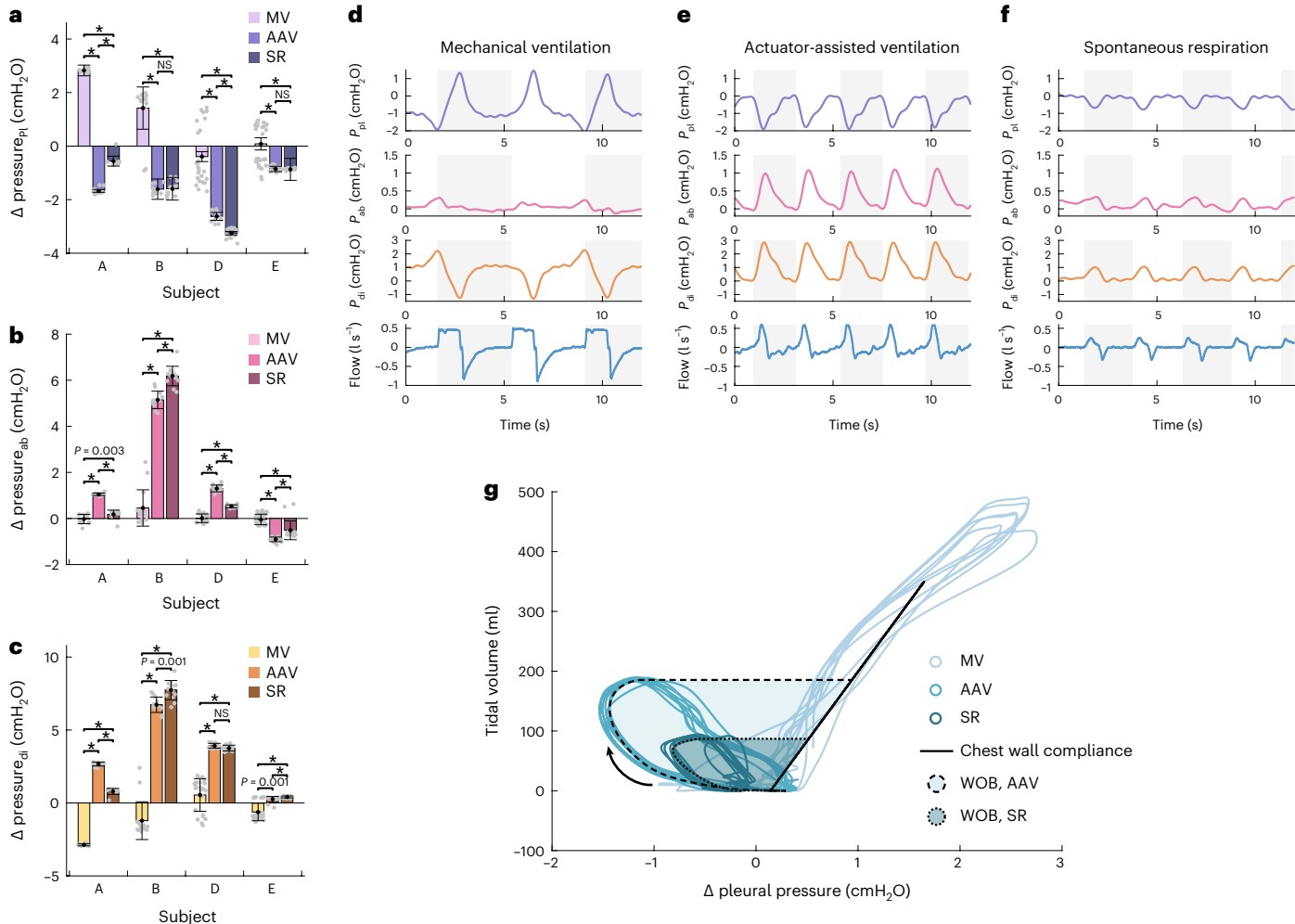

**Fig. 7 | Comparison of respiratory waveforms. a–c,** Average change in $P_{pl}$ (**a**), $P_{ab}$ (**b**) and $P_{di}$ (**c**) per breath under mechanical ventilation (MV), actuator-assisted ventilation (AAV) and spontaneous respiration (SR) taken from a representative steady-state segment from one respiratory challenge for 4 independent biological replicates (subjects A, B, D and E, with 11–32 breaths per subject). Each grey dot on the plots represents technical replicates within the subjects. **d–f,** Representative $P_{pl}$, $P_{ab}$, $P_{di}$ and flow waveforms for mechanical ventilation (**d**), actuator-assisted ventilation (**e**) and unassisted spontaneous respiration (**f**) from one respiratory challenge from 1 biological replicate. The alternating grey and white background indicates the bounds of each breath. **g,** Respiratory Campbell diagram plotting the pleural pressure-volume loops for representative breaths from MV, AAV and SR for 1 biological replicate. The direction of inspiration is indicated by the arrow. The compliance of the passive chest wall derived from the MV is indicated via the solid black line. The shaded regions outlined by the dashed/dotted lines indicate the area representative of the WOB. In **a–c,** bar plots and error bars show the mean ± s.d., *$P < 0.001$ using a two-sided Wilcoxon rank-sum test.

volume and inspiratory flow), biomechanics (cavity pressures, WOB), motion (ultrasonography and fluoroscopy) and gas exchange (ABGs).

## Contributions

The diaphragm-assist system generated substantial augmentation in respiratory function—measured via peak inspiratory flow (a direct metric of inspiratory function), and tidal volume and minute ventilation (metrics of ventilation)—in our most responsive subject. Subject A had the highest change in peak inspiratory pressure, tidal volume and minute ventilation; the corresponding large augmentation in peak inspiratory pressure indicates that the volume and minute ventilation augmentations are specifically due to the soft robotic actuators augmenting the diaphragm's inspiratory function. Responsiveness to the system varied across subjects.

Variance in responsiveness is probably dependent on a combination of many factors. One factor is the level of preserved respiratory baseline. The weak response in the subject with a relatively high preserved weight-normalized minute ventilation (subject E) suggests that the assist system may have weak augmentation or even a disruptive

effect in cases of well-preserved diaphragm function. Other potential factors include precise actuator placement, actuator fit and anatomical variations.

We showed that synchronization with the native respiratory effort is a critical design element in our system. Synchronous actuation is key to consistent, low-variance respiratory waveforms and tidal volumes. As in standard mechanical ventilation, off-cycle actuation of the actuators can lead to a destructive interference with the underlying respiratory effort, resulting in a poor augmentation and poor blood acid-base balance. In evaluating the effect of synchronization on the system's ability to maintain appropriate gas exchange, we demonstrated that despite generating a similar range of tidal volumes, independent actuation led to an inability to maintain appropriate $p_{CO2}$ levels and resulted in respiratory acidosis. Contrastingly, in two trials of well-synchronized actuation, we observed some capacity of the device to maintain and recover baseline $p_{CO2}$ levels.

The control system used in this study was a simple but effective first-generation system with many directions for improvement. The synchronization was triggered from airway flow, which is also the metric

used by gold standard clinical ventilatory support options for triggering, but flow is also the most downstream signal in neuro-ventilatory coupling. The downstream nature of the signal is a potential source of delays and asynchrony[34]. To achieve consistent assistance from breath to breath, the synchronization must be optimized for the alignment that maximizes constructive interference. The system relied on a manually titrated threshold set for the flow sensor data. It is designed to be triggered at the start of an inspiratory flow effort, which is related to $V_0$. However, the manual nature of the system meant that if the threshold was set too low, noise in the flow signal could cause pre-emptive or false triggering (as evidenced by the negative values for $P_0$-$V_0$). Our alignment analysis reveals two important considerations for improvement towards this goal. The first consideration is that the influence of alignment changes with the degree of preserved respiratory function, as seen with the difference in results between the intact and the severed phrenic nerve. When the phrenic nerve is severed, all diaphragm motion is governed by the actuators, and misaligned actuation with the remaining native respiratory effort—expansion of the ribcage—results in more consequential destructive interference. However, when the phrenic nerve is intact, the net diaphragm motion results from a combination of native diaphragm function and the effect of the actuators, because the actuators only operate along 2 discrete lines on the diaphragm. The contraction of the rest of the native diaphragm motion is still synchronized with the ribcage motion, so the effects of misalignment are less apparent. This implies that optimal alignment parameters may be different for different disease states and the control system will need to be dynamic and adaptive to changes in respiratory function, even within the same patient. The second consideration is that the actuation curve's relationship to the beginning of expiration ($V_{pk}$) is more influential than the relationship to the beginning of inspiration ($V_0$). This implies that an updated system should trigger from a signal related to expiration as opposed to the beginning of inspiration. Some neuromuscular signals, such as the electrical activity of the diaphragm (Edi), contain detailed information about both inspiration and expiration times[35,36]. Edi amplitude is also proportional to the neural drive, as well as the degree of contraction of the diaphragmatic muscle, therefore opening up the possibility of adaptive control. Triggering from Edi measured at the oesophageal level via a feeding tube[37] may be warranted to improve mechanical ventilation. This method, known as neurally adjusted ventilatory assist, is available in the clinical setting with mechanical ventilation and may improve respiratory weaning of patients that are challenging to wean[36]. The same principle could be applied to our diaphragm-assist system; using a more upstream signal with greater information on the native respiratory effort would allow for a more robust control system.

Overall, we show that the strategy to augment the native function of the diaphragm with soft robotics acts as a form of negative-pressure ventilation by driving ventilation through the generation of a negative pressure in the thoracic cavity. Our diaphragm-assist system is biomechanically similar to that of spontaneous breathing, sharing a substantial portion of the work of breathing in our best-responding subject. By functioning as an assist device, as opposed to completely taking over breathing, our system has the potential to be compatible with voluntary use of the diaphragm. Manoeuvres such as voluntary deep breaths or drinking through a straw—abilities related to patient autonomy and quality of life—can be preserved with this implantable ventilator strategy. Additionally, in contrast to current modes of mechanical ventilation, recapitulation of native biomechanics, as shown with this system, can avoid the deleterious effects that arise secondary to the use of positive pressure ventilation, such as barotrauma[38,39] or haemodynamic changes in patients with concurrent cardiac pathologies[40,41].

## Overall limitations

In this study, we demonstrate the foundational work towards a soft robotic implantable ventilator. Translationally, there are many hurdles to overcome between the proof-of-concept state presented here and the ultimately envisioned system, and we discuss them in the subsequent text.

Given that we saw variable responsiveness to the device across subjects, additional studies are needed to understand what factors in system design and implantation can replicate high responsiveness. Our system could generate the low end of acceptable minute ventilations but relied on high respiratory rates to do so. Given the presence of dead space, low tidal volumes result in less alveolar ventilation than if the same minute ventilation is achieved with higher tidal volumes and a lower respiratory rate. A core goal of the next-generation system is to further improve the tidal volume augmentation, which will need to be achieved through both actuator design and control system development.

Here we used the classic McKibben actuator; a more application-specific or customized actuator type may allow for further increases in tidal volumes in future work. Other factors in actuator design, such as the number, layout and positioning of actuators, will also be critical. We demonstrated tunability of assist by controlling pressurization, but an updated design will require finer characterization. Synchronization is critical to device performance, and thus future work lies in building a next-generation control system; this includes creating a system that is cognizant of the beginning of expiration as opposed to inspiration, an automated control system that removes the error of manual titration and further investigation of dynamic actuation curves. An ideal next-generation control system should aim to trigger from a more upstream neural signal, such as the electrical activity of the diaphragm, to provide an earlier signal that enables an advanced control system to optimize synchronization, removing delays and asynchrony. Neural triggering via implanted electrodes would also untether the current system from the flow instrumentation, freeing the patient from interventions at the mouth or trachea. To fully realize untethering from bulky machines, as in standard mechanical ventilators, the external components that control and power the system require miniaturization. Future work will aim to eventually miniaturize the system to the scale of a small backpack—one that could be worn by the patient or attached to a belt or an electric wheelchair. The process of miniaturization and portability has proved to be possible in similar complex devices, such as ventricular-assist devices (for example, Thoratec HeartMate III) or total artificial hearts (for example, Syncardia TAH, Carmat Aeson)[42–45].

## Towards clinical translation

Envisioning translation to the clinical field, the following considerations might help to optimize management and pave the way to human application. The diseases leading to chronic diaphragmatic dysfunction are numerous and feature very different pathophysiologies. Therefore, a thorough understanding of the underlying pathology as well as its specificity is critically needed to help optimize management and anticipate complications[46]. Moreover, patient selection and indication will need to be clearly defined, to select patients who will benefit from this therapy the most. Here we present a generalized mechanical strategy for diaphragm support, but the parameters of actuator design or actuation control will need to be optimized and specialized on the basis of the needs of a given pathology as well as individual patient anatomy.

Owing to the complexity of the procedure, a multidisciplinary team highly trained in advanced thoracic surgery is required to build expertise and develop this technology, ideally in a high-volume centre[47]. Technological improvement is required to provide the least invasive approach of implantation. In this regard, a thoracoscopic route might be beneficial and will be the subject of future work. Given the invasive nature of implantable devices, the diaphragm-assist system is targeted towards patients with chronic-to-permanent ventilator dependence. We recognize that surgery in patients suffering severe diaphragm dysfunction causing respiratory failure can carry a high morbidity and mortality. Peri-operative complications can be numerous; one of the

most feared is the worsening of the pulmonary status, which may itself precipitate the need for long-term ventilation[48]. Nevertheless, it has been well demonstrated that complex thoracic surgery is feasible even in very frail patients. Lung transplantation for terminal respiratory disease[49] is one of the most striking examples. Thus, surgery could still be considered in a suitable target population that would ultimately benefit from this mechanical augmentation of diaphragm function, such as a range of neuromuscular disorders. The concept of diaphragm assist is in itself a means of preventing further complications from chronic respiratory failure and preserving key aspects of quality of life, such as speech and mobility.

Owing to the focus on feasibility, we acknowledge that there are limitations in these acute studies from the lens of regulatory approval and clinical translation. We did not study device biocompatibility or long-term device operation. The device was constructed from types of polymers that are already used in established medical devices[50–53], such as poly(ethylene terephthalate) (PET) and polyurethanes (Supplementary Information). Because the device focuses on mechanical interaction, as opposed to biochemical interactions with the body, the materials used in the device can easily be substituted with regulatory-approved materials in future iterations. With improved performance and stability, future long-term studies will need to investigate the long-term effects of the system, including tissue remodelling and the ability to provide full-time respiratory support.

The technology requires further advancements in the net tidal volumes it can generate before it can fully match the ventilation capacity of a current mechanical ventilator. We envision the further translational potential of this technology when combined with the development of smaller and more portable pneumatic energy sources[54,55] as the field of soft robotics advances. With the integration of a portable pump and control system, the technology could provide an additional level of patient autonomy via increased mobility. We believe that with optimized design, the technology may provide a radically different ventilation technology that preserves key metrics of quality of life for people with end-stage mechanical respiratory failure.

## Methods

### Study design

There were two main objectives of our study. First, we sought to demonstrate the proof-of-concept capability to augment ventilation via implanted soft robotic actuators in an animal model of respiratory muscle weakness. To evaluate ventilation metrics, we measured spirometric flow and volume. Second, we aimed to demonstrate that this soft robotic strategy replicates more native respiratory biomechanics than standard mechanical ventilation. To evaluate the respiratory biomechanics, we evaluated the respiratory pressure data along with the spirometry data.

To evaluate system performance under varying conditions within a single animal, a series of respiratory challenges were performed. Before the first and between subsequent respiratory challenges, volume control mechanical ventilation operated through the facility's Drager Tiro ventilator (Drägerwerk) was used to maintain the animal's ventilation needs and recover from respiratory challenges if needed. Measurements of arterial blood gases were taken to validate normal baseline respiratory status before each challenge. Each respiratory challenge was initiated by switching the ventilator to a manual mode of ventilation. Data for a mix of unsupported ventilation and actuator-supported ventilation were collected. Vital signs and respiratory status were monitored. For experiments with uninterrupted trials, ABGs were collected at 2 or 5 min intervals during the challenge.

### Fabrication and characterization of PAM

The actuators were a modified version of previously described PAM actuators[7,9]. Specifically, McKibben pneumatic artificial muscles were fabricated according to the protocol detailed in Supplementary

Methods. Actuator dimensions were selected to fit the anatomical needs of the 30–40 kg swine. They consist of a thermoplastic elastomer bladder (Stretchlon 200, FibreGlast), a thermoplastic polyurethane tubing (1/8 inch tubing, 5648K226, McMaster) and an expandable braided mesh (PTO0.25BK, TechFlex). Before in vivo use, actuators were fatigue tested to a pressurization of 20 psi for >1,000 cycles on the benchtop. Mechanical characterization was performed on an Instron 5499 universal testing system (Instron).

Actuator characterization was conducted both in vitro and in vivo. For the in vitro characterization, actuator performance was measured via Instron testing. Classic tensile testing was conducted to measure the contractile force. A modified flexural bend setup (Supplementary Fig. 5) was used to measure the perpendicular force applied to the diaphragm via arclength shortening. For the in vivo characterization, performance of the diaphragm-assist system was evaluated through the diaphragm displacement (via ultrasonography) and the functional metrics (tidal volume, Campbell diagram) (Extended Data Figs. 1 and 2). Different pressurization shapes and levels were input into the actuator (Extended Data Figs. 1 and 2) and the resulting behaviour was measured. Further details can be found in Supplementary Information.

### Live animal studies

All studies were conducted according to protocol no. 19-05-3907 approved by the Boston Children's Hospital (BCH) Institutional Animal Care and Use Committee (IACUC) policy.

Procedures were carried out at Boston Children's Hospital in accordance with BCH IACUC under protocol no. 19-05-3907 and MIT IACUC under protocol no. 0118-006-21. Protocol reviews were conducted in accordance with the standards outlined in the National Research Council's Guide for the Care and Use of Laboratory Animals and BCH's Animal Welfare Assurance.

Female Yorkshire (30–40 kg) swine were sourced from Parson's Farm (Hadley, MA, USA). We used a total of 12 swine during the development and testing of our system, and we present data from 9 swine in the manuscript. Different subsets of subjects were used for the experimental investigations reported; not all subjects were used in every experimental investigation. Animals were acclimated and cared for according to standard facility protocols. Each experiment was conducted under 2–3% isoflurane anaesthesia, titrated to each animal to maintain a stable anaesthetic plane. Anaesthesia and mechanical ventilation were controlled through the facility's Drager Tiro ventilator (Drägerwerk). Vital signs were monitored via a SurgiVet monitor (Smiths Medical). After completing the study and acquiring the data, animals were euthanized using Fatal-Plus solution (Vortech Pharmaceuticals) at a dose of 110 mg kg$^{-1}$ body weight.

### Surgical procedure

After induction of anaesthesia, the animal was intubated and placed on mechanical ventilation. A trans-oesophageal electrocardiogram catheter was placed to monitor the heart rate. A carotid arterial sheath and jugular venous line were placed using cut-down technique for animal systemic and central venous pressures monitoring, respectively. Two balloons were placed, one in the oesophagus and one in the stomach, for pressure monitoring. A Foley catheter was placed for urine output monitoring.

Subsequently, the chest cavity was accessed through midline sternotomy. Next, we opened both pleural cavities and placed one soft actuator along the diaphragm curvature on each cavity. The anterior portion was attached to the sternum and the posterior attachment was made to the lowest posterior rib in the most medial position that can be achieved without disrupting the region of the major arteries and veins, oesophagus and spine. To do this, we passed each actuator posteriorly at lowest intercostal space to outside the chest cavity and fixed it to the skin using sutures. Then, we fixed the other end to the sternum using sutures and we passed the actuation lines through a separate opening through the skin. Next, we approximated the sternum using

sternal wires and closed the subcutaneous layers and the skin in layers using sutures. After the sternotomy was closed, the negative pressure in the thoracic cavity was restored via chest tube, and the respiratory challenges were conducted with a closed chest.

## Simulating varying levels of respiratory function

To simulate varying levels of respiratory functions, two animal models of respiratory muscle weakness were used. The first method relied on the respiratory depressive effects of isoflurane. Isoflurane levels were held between 2–3% and titrated to a stable plane of anaesthesia while still maintaining a depressed but non-zero level of spontaneous respiration during respiratory challenges. The second method modelled diaphragm paralysis by mechanically severing both the left and right phrenic nerves. This model was still conducted under the setting of the isoflurane, and thus combines the effects of the isoflurane and severed phrenic nerve, and represents a more severe model of respiratory weakness.

## Data acquisition

The biomedical sensors and instrumentation data were input into a PowerLab 35 series (PL3516, ADInstruments) high-performance data acquisition system with a 1,000 Hz sampling frequency for all channels. During the experiments, data were monitored live via LabChart software (ADInstruments). After the experiments, data were exported into and processed in MATLAB (MathWorks).

## Spirometry instrumentation

An analogue spirometer (Gas Flow Sensor, ES Systems) was placed in line between the ventilator Y-tubing and the endotracheal tube. Analogue data were input into PowerLab. The data were converted from mass flow to volumetric flow according to manufacturer specifications.

## Respiratory pressure instrumentation and measurement

Pleural pressure and abdominal pressure were measured via oesophageal balloon catheters (Cooper Surgical) placed in the oesophagus and stomach, respectively, each connected to a pressure transducer (PRESS-S-000, PendoTech).

Respiratory pressure data were normalized in MATLAB post processing. For a given segment of interest, the average of the pressure reading at the breath bounds was set to zero to allow the analysis to show the change in pressure over the course of one breath.

## Ultrasonography

Ultrasonography, a non-invasive, non-ionizing imaging method, was used to investigate the interactions of the device with the diaphragm. Ultrasonography can be used to assess diaphragm displacement and dysfunction[25]. More precisely, it allows direct two-dimensional visualization of the diaphragm, permitting quantification of its motion and function, and serves as an ideal tool to assess the interaction of the device with the diaphragm. A Philips iE33 (Philips Healthcare) echography machine was used with the X7-2 transducer (Philips Healthcare). A two-dimensional image (so-called B-mode, Brightness) of the diaphragm and the device was obtained by placing the probe in the right subcostal space, pointing in the cranial direction. To quantify the motion of the device and the diaphragm, M-mode was used.

## Control system design and instrumentation

Our group has built a custom electropneumatic control system utilizing electropneumatic pressure regulators and valves (SMC Pneumatics, SMC) controlled by a custom software described in ref. [56]. The software is designed to allow custom pressure waveforms to be input. The control system can generate a desired waveform via an analogue input to the electropneumatic regulators. The nominal peak pressure for all waveforms was 20 psi. The regulators also output an analogue signal of the actual pressure waveform; these data were input into the PowerLab system.

## Independent and synchronized actuation

The custom control system can generate a manual timing set to a frequency of actuation that is initiated by user input. This set timing initiates the custom pressure waveform programmed into the system and is independent of the subject's native breathing.

To implement synchronization in our system, we used the Fast Response Output add-on for LabChart (ADInstruments). Analogue spirometry flow data were used as the input channel. Voltage and hysteresis settings were manually titrated between a voltage range equivalent to $0.01 \, l \, s^{-1}$ to $0.07 \, l \, s^{-1}$ and a hysteresis range of 2–5% during every respiratory trial to achieve qualitatively good synchronization, as visually recognized by the homogeneity of the real-time flow and volume waveforms. The digital output channel on the PowerLab system was used to send a trigger pulse to a digital input channel in the microcontroller of the custom control system described above.

## Statistical analysis

Statistical tests were conducted as described in the respective figure captions for Figs. 3, 4 and 7 and Extended Data Figs. 1 and 2. For Figs. 3c,d and 7a–c, two-sided Wilcoxon rank-sum analyses were conducted in MATLAB (MathWorks) via the 'ranksum' function. Fig. 4e,f depicts two sets of statistical tests. A two-sided Welch's $t$-test without assuming equal variances was conducted to compare the means of the populations via the 'ttest2' function in MATLAB with an 'unequal' variance type specification. Additionally, a two-sample $F$-test for equal variances was conducted to compare and confirm unequal variances via the 'vartest2' function in MATLAB. For Extended Data Figs. 1 and 2, two-sided $t$-tests were conducted via the 'ttest2' function in MATLAB. Significance denoted in figures is $*P < 0.001$, unless an exact $P$ value is given.

## Reporting summary

Further information on research design is available in the Nature Portfolio Reporting Summary linked to this article.

## Data availability

The main data supporting the findings of this study are available within the Article and its Supplementary Information. Additional data are available from the corresponding author on request. Source data for the figures are provided with this paper.

## Code availability

The custom MATLAB codes used in this study are available at https://github.com/RocheLab/ImplantableVentilator.

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

## Acknowledgements

L.H. discloses support for the research described in this study from CIHR Skin Research Training Centre (201710DFS) and Muscular Dystrophy Association (577961). L.H., M.Y.S, M.S., D.Q.M. and E.T.R. disclose support for the publication of this study from the National Institutes of Health (NIH), National Institute of Biomedical Imaging and Bioengineering (NIBIB), grant R21-EB028414-01A1. J.B. discloses support from the SICPA Foundation and Lausanne University Hospital Improvement fund. D.Q.M. acknowledges the SMA2 Brown Fellowship, Massachusetts Institute of Technology. E.T.R. discloses support from National Science Foundation (NSF) grant 1847541.

## Author contributions

L.H. and E.T.R. conceived the hypothesis. L.H., E.T.R., J.B. and N.V.V. designed the experiment. L.H., J.B., M.Y.S., M.S. and D.Q.M. performed the experiments. L.H., J.B., M.S. and E.T.R. analysed the results. L.H., J.B., M.S. and E.T.R. wrote the manuscript.

## Competing interests

The authors declare no competing interests.

## Additional information

**Extended data** is available for this paper at https://doi.org/10.1038/s41551-022-00971-6.

**Correspondence and requests for materials** should be addressed to Ellen T. Roche.

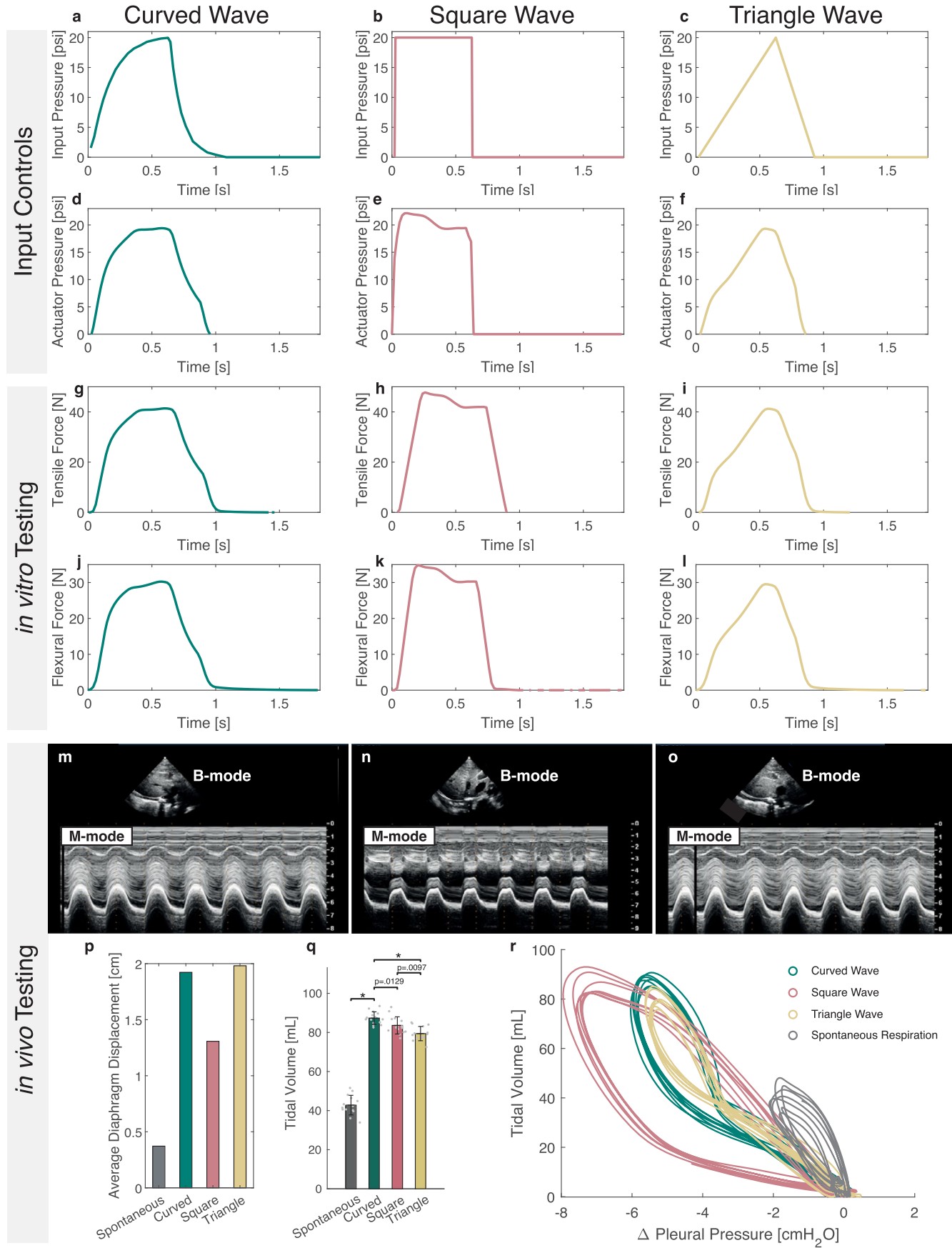

**Extended Data Fig. 1 | See next page for caption.**

**Extended Data Fig. 1 | Controlling actuation via different pneumatic waveforms.** Input waveforms of a (**a**) curved, (**b**) square, and (**c**) triangle shape can be programmed into the custom-built control system. The effective output pressure of the electropneumatic regulator for the (**d**) curved, (**e**) square, and (**f**) triangle shape drives actuation. The PAM actuation forces were characterized for different waveforms *in vitro* on a classic Instron tensile test setup (**g,h,i**) and our modified flexural test setup (**j,k,l**) (depicted in Supplementary Fig. 5). Input waveforms of a (**m**) curved, (**n**) square, and (**o**) triangle shape generate different shapes of diaphragm displacement as visualized via M-mode ultrasound. **p**, Average diaphragm displacement from (**m,n,o**). **q**, Average tidal volume and (**r**) respiratory Campbell diagram plotting the pleural pressure-volume loops for representative breaths from different waveform shapes. (**m-r**) represent one biological replicate. In (**q**), bar plot shows mean, error bars ±s.d., *p < 0.001 using a two-sided t-test. Each grey dot represents a technical replicate (14–15 breaths per bar).

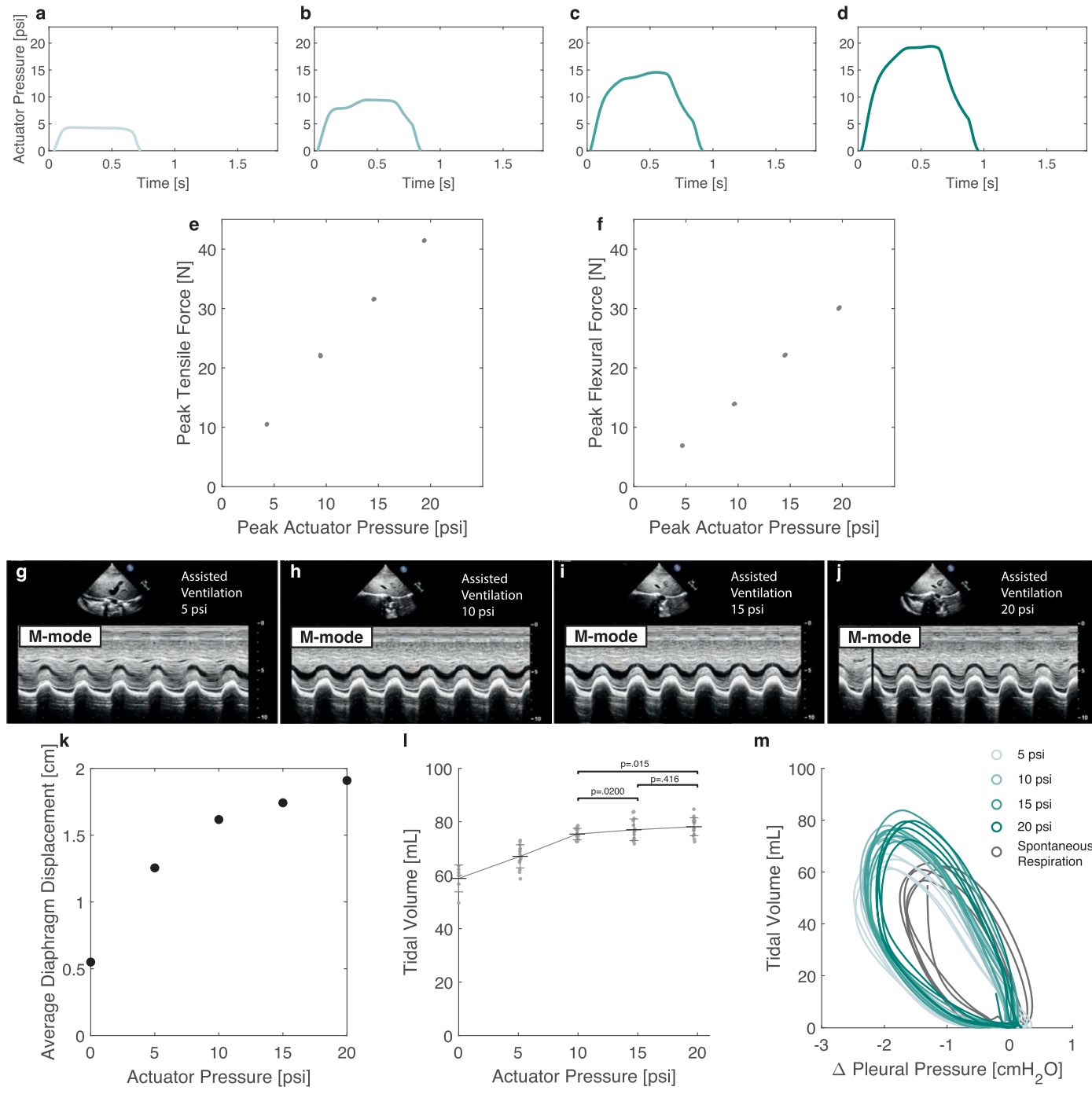

**Extended Data Fig. 2 | Tuning actuation depth via level of pressurization.**
The actuator pressure profile for a curved waveform scaled to have a peak
nominal pressure of (**a**) 5 psi, (**b**) 10 psi, (**c**) 15 psi, (**d**) 20 psi. The peak forced
generated by different levels of actuation were characterized in vitro on a
(**e**) classic Instron tensile test setup and (**f**) our modified flexural test setup
(depicted in Supplementary Fig. 5). Diaphragm displacement generated by
actuations of (**g**) 5 psi, (**h**) 10 psi, (**i**) 15 psi, (**j**) 20 psi visualized via M-mode
ultrasound. (**k**) The average diaphragm displacement per breath from one

sample subject via M-mode ultrasound. (**l**) Tidal volume achieved via
different levels of pressurization from one sample subject. Significance is
indicated by p values using a two-sided t-test. Error bars show ±s.d. Each grey
dot represents a technical replicate (6–15 breaths per level of pressurization).
(**m**) Respiratory Campbell diagram plotting the pleural pressure-volume loops
for representative breaths from different levels of actuation. (**g-m**) represent
one biological replicate.

# Reporting Summary

## Statistics

For all statistical analyses, confirm that the following items are present in the figure legend, table legend, main text, or Methods section.

| n/a | Confirmed | |
|---|---|---|
| ☐ | ☒ | The exact sample size (*n*) for each experimental group/condition, given as a discrete number and unit of measurement |
| ☐ | ☒ | A statement on whether measurements were taken from distinct samples or whether the same sample was measured repeatedly |
| ☐ | ☒ | The statistical test(s) used AND whether they are one- or two-sided<br>*Only common tests should be described solely by name; describe more complex techniques in the Methods section.* |
| ☒ | ☐ | A description of all covariates tested |
| ☒ | ☐ | A description of any assumptions or corrections, such as tests of normality and adjustment for multiple comparisons |
| ☐ | ☒ | A full description of the statistical parameters including central tendency (e.g. means) or other basic estimates (e.g. regression coefficient) AND variation (e.g. standard deviation) or associated estimates of uncertainty (e.g. confidence intervals) |
| ☐ | ☒ | For null hypothesis testing, the test statistic (e.g. *F*, *t*, *r*) with confidence intervals, effect sizes, degrees of freedom and *P* value noted<br>*Give P values as exact values whenever suitable.* |
| ☒ | ☐ | For Bayesian analysis, information on the choice of priors and Markov chain Monte Carlo settings |
| ☒ | ☐ | For hierarchical and complex designs, identification of the appropriate level for tests and full reporting of outcomes |
| ☒ | ☐ | Estimates of effect sizes (e.g. Cohen's *d*, Pearson's *r*), indicating how they were calculated |

*Our web collection on statistics for biologists contains articles on many of the points above.*

## Software and code

Policy information about availability of computer code

| Data collection | LabChart 8 software. |
|---|---|
| Data analysis | Data from LabChart 8 was exported to MATLAB R2021a. Data analysis was conducted via custom code written in MATLAB R2021a, to analyse parameters such as peak inspiratory flow and tidal volumes. The custom code is available at https://github.com/RocheLab/ImplantableVentilator. |

For manuscripts utilizing custom algorithms or software that are central to the research but not yet described in published literature, software must be made available to editors and reviewers. We strongly encourage code deposition in a community repository (e.g. GitHub). See the Nature Portfolio guidelines for submitting code & software for further information.

## Data

Policy information about availability of data

All manuscripts must include a data availability statement. This statement should provide the following information, where applicable:
- Accession codes, unique identifiers, or web links for publicly available datasets
- A description of any restrictions on data availability
- For clinical datasets or third party data, please ensure that the statement adheres to our policy

The main data supporting the findings of this study are available within the article and its Supplementary Information. Source data for the figures are provided with this paper. Additional data are available from the corresponding author on request.

# Field-specific reporting

Please select the one below that is the best fit for your research. If you are not sure, read the appropriate sections before making your selection.

☒ Life sciences ☐ Behavioural & social sciences ☐ Ecological, evolutionary & environmental sciences

For a reference copy of the document with all sections, see nature.com/documents/nr-reporting-summary-flat.pdf

# Life sciences study design

All studies must disclose on these points even when the disclosure is negative.

| | |
|---|---|
| Sample size | We used a total of twelve swine during the development and testing of our system, and we present data from nine swine in the paper. For each subject, a series of respiratory challenges and conditions were tested. Different subsets of subjects were used for the experimental investigations reported; not all subjects were used in every experimental investigation. In this study, each breath provides a data point for the various analyses shown. A single respiratory challenge supplies a large pool of data. |
| Data exclusions | Data from nine swine are presented in this paper (six for quantitative data, and three for echocardiographic measurements). The data from the three swine not shown were not conducted under the same conditions, and therefore are not presented.<br><br>For Fig. 3, panels a and b are representative datasets. In this figure, the six swine presented are labeled A–F. Five of six swine (subjects A–E) are shown in Fig. 3c,d; subject F did not have data for the apnea condition at the start of the respiratory challenge and is therefore not shown.<br><br>Fig. 6 depicts the analysis of aligning the actuator synchronization to the underlying respiratory effort for two respiratory challenges within subject B.<br><br>Fig. 7 a–c omits subject C and F because the pressure-sensing instrumentation was nonfunctional for those two swine and the data for this parameter was not collected. |
| Replication | The trends shown in Fig. 6 are from one respiratory challenge within the animal. To ensure replication, this analysis was conducted on all data from this subject, and the reported trend replicates in a different respiratory challenge within the same animal.<br><br>For other subjects, the synchronization was actually overly consistent, and therefore lacked the variability in timing needed to visualize these trends. In order to replicate these results in another subject, a sweep of the synchronization/delay will generate the appropriate variability needed. |
| Randomization | The full set of respiratory challenges were conducted in each of 6 subjects, but in a semi-random order. The order of the independent and synchronous challenges was randomized, but the challenges conducted with a severed phrenic nerve necessarily came after all of the data with an intact phrenic nerve were collected. Echocardiography was performed in three swine. |
| Blinding | Individual subjects were not grouped and instead a variety of respiratory challenges were conducted within each subject. |

# Reporting for specific materials, systems and methods

We require information from authors about some types of materials, experimental systems and methods used in many studies. Here, indicate whether each material, system or method listed is relevant to your study. If you are not sure if a list item applies to your research, read the appropriate section before selecting a response.

## Materials & experimental systems

| n/a | Involved in the study |
|---|---|
| ☒ | ☐ Antibodies |
| ☒ | ☐ Eukaryotic cell lines |
| ☒ | ☐ Palaeontology and archaeology |
| ☐ | ☒ Animals and other organisms |
| ☒ | ☐ Human research participants |
| ☒ | ☐ Clinical data |
| ☒ | ☐ Dual use research of concern |

## Methods

| n/a | Involved in the study |
|---|---|
| ☒ | ☐ ChIP-seq |
| ☒ | ☐ Flow cytometry |
| ☒ | ☐ MRI-based neuroimaging |

## Animals and other organisms

Policy information about studies involving animals; ARRIVE guidelines recommended for reporting animal research

| | |
|---|---|
| Laboratory animals | Yorkshire swine, female, 30–40kg. |
| Wild animals | The study did not involve wild animals. |

| Field-collected samples | The study did not involve samples collected from the field. |
|---|---|
| Ethics oversight | All studies were conducted according to protocol #19-05-3907, approved by the Boston Children's Hospital (BCH) Institutional Animal Care and Use Committee (IACUC; policy and MIT protocol 0121-001-23). |

Note that full information on the approval of the study protocol must also be provided in the manuscript.

