## [Peer Review File · Nature Biomedical Engineering]

An implantable soft robotic ventilator augments inspiration in a pig model of respiratory insufficiency

Corresponding author: Ellen Roche

Editorial note

This document includes relevant written communications between the manuscript's corresponding author and the editor and reviewers of the manuscript during peer review. It includes decision letters relaying any editorial points and peer-review reports, and the authors' replies to these (under 'Rebuttal' headings). The editorial decisions are signed by the manuscript's handling editor, yet the editorial team and ultimately the journal's Chief Editor share responsibility for all decisions.

Any relevant documents attached to the decision letters are referred to as **Appendix #**, and can be found appended to this document. Any information deemed confidential has been redacted or removed. Earlier versions of the manuscript are not published, yet the originally submitted version may be available as a preprint. Because of editorial edits and changes during peer review, the published title of the paper and the title mentioned in below correspondence may differ.

Correspondence

Mon 04 Apr 2022

Decision on Article nBME-21-2902

Dear Prof Roche,

Thank you again for submitting to *Nature Biomedical Engineering* your manuscript, "An implantable ventilator augments inspiration in an *in vivo* porcine model". As noted in previous e-mail correspondence, the manuscript has been seen by three experts, yet despite our chasing efforts, one reviewer has failed to provide a report. The feedback from two reviewers, which I had already forwarded to you, is also included at the end of this message.

You will see that the reviewers appreciate the work. However, they express concerns about the degree of support for the claims, and provide useful suggestions for improvement. We hope that with significant further work you can address the criticisms and convince the reviewers of the merits of the study. In particular, we would expect that a revised version of the manuscript addresses the concerns on the safety and performance of the implantable ventilator, and includes additional caveats as to the expected bottlenecks that would need to be overcome before eventual human trials of the device. Importantly, please make sure that device characterization and methodology reporting are thorough.

When you are ready to resubmit your manuscript, please upload the revised files, a point-by-point rebuttal to the comments from all reviewers, the reporting summary, and a cover letter that explains the main improvements included in the revision and responds to any points highlighted in this decision.

Please follow the following recommendations:

* Clearly highlight any amendments to the text and figures to help the reviewers and editors find and understand the changes (yet keep in mind that excessive marking can hinder readability).- * If you and your co-authors disagree with a criticism, provide the arguments to the reviewer (optionally, indicate the relevant points in the cover letter).
- * If a criticism or suggestion is not addressed, please indicate so in the rebuttal to the reviewer comments and explain the reason(s).
- * Consider including responses to any criticisms raised by more than one reviewer at the beginning of the rebuttal, in a section addressed to all reviewers.
- * The rebuttal should include the reviewer comments in point-by-point format (please note that we provide all reviewers with the reports as they appear at the end of this message).
- * Provide the rebuttal to the reviewer comments and the cover letter as separate files.

We hope that you will be able to resubmit the manuscript within 20 weeks from the receipt of this message. If this is the case, you will be protected against potential scooping. Otherwise, we will be happy to consider a revised manuscript as long as the significance of the work is not compromised by work published elsewhere or accepted for publication at *Nature Biomedical Engineering*.

We hope that you will find the referee reports helpful when revising the work. Please do not hesitate to contact me should you have any questions.

Best wishes,

Pep

Pep Pàmies
Chief Editor, Nature Biomedical Engineering

Reviewer #2 (Report for the authors (Required)):

This paper is the first demonstration of an in vivo implanted system for assisting in breathing and for contracting diaphragm. The use of external ventilators is very critical, and having an implanted system would solve major problems for patients with diaphragm dysfunctions.

I appreciated the paper: the motivations are clear, the in vivo tests are sufficiently solid (even if quite short) and the future impact could be dramatic.

On the other hand, I have identified some major and minor issues which prevent me to give a positive assessment.

1. First of all, the engineering part is quite limited. All the used components are not optimized for the application. Soft actuators design, size and distribution are not considered in the study protocol (see also comment 4). Sensors and controllers are commercial. The most relevant parts of the paper are the animal test and the physiological measurements.
2. The clinical motivation of this work is very clearly presented at the beginning. The authors are invited to discuss (possibly in the discussion section) if the target patients for this kind of device are eligible to undergo such complex implant surgery procedure. This might be crucial to foresee future clinical employments of the device.
3. In the introduction section the authors should better contextualize their work not only with reference to soft actuators but also to fully implantable robots. Few examples and field definition were recently reported in the state of the art. Just as examples, please refer to: - Damian, D. D., Price, K., Arabagi, S., Berra, I., Machaidze, Z., Manjila, S., ... & Dupont, P. E. (2018). In vivo tissue regeneration with robotic implants. *Science Robotics*, 3(14). - Iacovacci, V., Tamadon, I., Kauffmann, E. F., Pane, S., Simoni, V., Marziale, L., ...

& Menciassi, A. (2021). A fully implantable device for intraperitoneal drug delivery refilled by ingestible capsules. *Science Robotics*, 6(57), eabh3328. - Menciassi, A., & Iacovacci, V. (2020). Implantable biorobotic organs. *APL bioengineering*, 4(4), 040402.

4. It is not clear how the number of PAMs and the overall output force required were set. Please clarify on this and comment on the role played by spatial distribution of the two actuators on the diaphragm surface. What about the needed PAM contraction? Should this be adjusted for each subject and then be constant over the respiratory task?

5. The authors employed a custom controller aimed at synchronizing the soft robotic actuator and natural breathing. However, both sensing (spirometry) and control rely on external (bulky?) data acquisition and control platforms. This appears a bit in contrast with the original motivation to outperform over invasive ventilation systems. No concrete paths towards implantations are given.

6. The caption of figure 2 is very informative. However, it would be useful to add some labels on the axes, graphs title, etc., to make figures readability more straightforward.

7. The in vivo test is very short, so all considerations about materials, biocompatibility, stability etc. are missing.

8. In line 512, 6 pigs are mentioned. But in the rest of the papers we had 5 pigs.

Reviewer #3 (Report for the authors (Required)):

This work by Drs. Hu and Roche developed an implantable device that can substitute a mechanical ventilator commonly used in the contemporary clinical care and yet, avoid the various shortcomings of it. The authors build a robotic actuator which is overlaid to the subject's diaphragm and activated by the host's inspiratory effort, which in turn, is balloon inflated and substitutes the function of the native diaphragm.

The authors challenge an important clinical question, with a good understanding of the respiratory cycle and the role of the diaphragmatic motion. The concept underlying the current work is interesting and innovative. I acknowledge that this is a prototype, however the current manuscript do have room for improvement. Additionally, considering the invasiveness of the implantation associated with their device, whether the strength of the system is enough to overcome the invasiveness associated with the procedure should be discussed more in depth.

Here are some major comments for the authors.

1. The authors fail to analyze the actual blood gas levels with arterial blood. The authors state that the responsiveness of the diaphragm can vary between subjects. How can the authors be sure that the blood gas levels are maintained at a relatively stable state with the device instillation? Please be aware that maintaining the blood acid-base balance in a homeostatic state is a critical role of respiration in every living creature.

2. How is the level of inspiration controlled with their device? The degree of inspiration should be controllable according to the subjects needs in various clinical situation (for example, in early sepsis, it is natural to be in a hyperventilated state) but the authors do not show that their device is versatile enough to accommodate this. What happens to the subject's respiration if the bladder in the system is pressurized more? Additionally, what about the opposite where there is no need for a deep inspiration?

3. Likewise, I would like to see how the respiratory mechanics change according to how fast the bladder in their system is inflated. More information should be given on this. The respiratory mechanics are shown as simple bar graphs in Figures 5a, b and c, whereas the gradient or slope of the Campbell diagram may also be important.

4. The authors should be commended for doing a nice job in synchronizing the device to the subject's respiratory effort. However, the trigger for this is the spirometer that is installed into the subject's airway. To really overcome the weakness of the contemporary mechanical ventilators, the authors should think of better methods that do not touch the respiratory tract to sense the intrinsic respiratory effort.

There are also some minor comments that might help.

1. Supplementary Video Online is not that informative as it plays in a real-time. It would be better if the video is played more slowly.

2. The finding should be supported by adequate statistical analysis.

2.1. Although somewhat obvious from the graphs, there should be some statistical analysis to show the difference in the variation of peak inspiratory flow and tidal volume of Figure 3e and f.

2.2. The pairwise comparisons should be done for Figure 5a, b, and c with the data acquired at the spontaneous respiration (SR) as the reference, not the actuator assisted ventilation (AAV).

2.3. How many samples were taken for each subjects in each graph. This should be noted.

3. In Figures 2c, d and e, when did the authors gather data after the device was turned off or on? The same questions goes for synchronized actuation in Figures 3e and f, Figures 5a, b and c.

4. Figure 4a and b miss the legends for y-axis.

5. The overall structure of the Discussion could be improved. The contents of each paragraphs tend to jump, making it difficult to follow.

Sun 16 Oct 2022

Decision on Article nBME-21-2902A

Dear Prof Roche,

Thank you for your patience in waiting for the guidelines for the final submission of your manuscript, "An implantable ventilator augments inspiration in an in vivo porcine model" to *Nature Biomedical Engineering*. Please carefully follow the instructions provided in the attached file.

Also, please consider the minor points from Reviewer #3.

For primary research originally submitted after December 1, 2019, we encourage authors to take up transparent peer review. If you are eligible and opt in to transparent peer review, we will publish, as a single supplementary file, all the reviewer comments for all the versions of the manuscript, your rebuttal letters, and the editorial decision letters. **When submitting the final version of your manuscript please indicate whether you opt in to transparent peer review.** In the interest of confidentiality, we allow redactions to the rebuttal letters and to the reviewer comments. If you are concerned about the release of confidential data, please indicate in the cover letter what specific information you would like to have removed; we cannot incorporate redactions for any other reasons. More information on transparent peer review is available.

When you are ready to submit the final version of your manuscript, please upload the files specified in the instructions file.

Best regards,

Pep

Pep Pàmies
Chief Editor, Nature Biomedical Engineering

Reviewer #2:

Report for the authors (Required):

The paper has been substantially improved and now the contribution of the work, the methodology behind and the main limitations of the obtained results have been reported. I think these additional parts (and additional tests, which I really appreciate) can help to position the paper in the state of the art and help the readers to understand the most critical elements for further developments. I am sure that some activities reported as "future steps" are not trivial at all (miniaturization of the electronics, biocompatibility, neural triggering), but the results, as a first proof of concept in vivo, deserve to be published.

Reviewer #3:

Report for the authors (Required):

This report by Drs. Hu and Roche is a revised version of the manuscript previously submitted to the Nature BME. The manuscript has been significantly improved and much of the parts have been clarified. A significant proportion of the previous questions have been resolved. I have two minor comments/suggestions remaining.

1. The blood gas analysis allows us to see how the system developed by the authors translate into the respiration of a given subject/animal but there are some comments related to this.

1.1. I am actually not sure whether the acidosis seen in Fig. 5b is a result of metabolic acidosis. Although the

blood gas results may be different between humans and swines, I think the blood gas analysis in Fig. 5b is a mixed form of both respiratory and metabolic acidosis.

1.2. How many animals were analyzed for the blood gas?

2. Although I agree that there is a lot to discuss, the length of the Discussion is actually quite long.

Nature Biomedical Engineering is a Transformative Journal. Authors may publish their research with us through the traditional subscription-access route, or make their paper immediately open access through payment of an article-processing charge. More information about publication options is available.

You may need to take specific actions to comply with funder and institutional open-access mandates. If the work described in the accepted manuscript is supported by a funder that requires immediate open access (as outlined, for example, by Plan S) and your manuscript was originally submitted on or after January 1st 2021, then you will need to select the gold OA route. Authors selecting subscription publication will need to accept our standard licensing terms (including our self-archiving policies), and these will supersede any other terms that the author or any third party may assert apply to any version of the manuscript.

Rebuttal 1

Manuscript nBME-21-2902 : Response to Reviewers' Comments

We thank the reviewers for their thoughtful and detailed feedback. We have addressed each of these comments in turn, and have included corresponding modifications to the text. We feel the reviewer's comments and the subsequent revisions have substantially clarified and strengthened the manuscript. For clarity, the changes made to the manuscript—including the new Extended Data—are shown in blue in both this response and in the manuscript. Changes made to the Supplemental Information are shown in orange in this document.

Comments from Editor

You will see that the reviewers appreciate the work. However, they express concerns about the degree of support for the claims, and provide useful suggestions for improvement. We hope that with significant further work you can address the criticisms and convince the reviewers of the merits of the study. In particular, we would expect that a revised version of the manuscript addresses the concerns on the safety and performance of the implantable ventilator, and includes additional caveats as to the expected bottlenecks that would need to be overcome before eventual human trials of the device. Importantly, please make sure that device characterization and methodology reporting are thorough.

Answer to the Editor

We thank the Editor for considering our manuscript for Nature Biomedical Engineering. Here, we present additional data and discussion to address the reviewer comments on safety and performance. Overall, we have conducted additional large animal experiments to better characterize our device and evaluate its performance. We investigated different actuation regimes and introduced two new metrics of performance: ultrasonography (seen in Fig. 2, 4, and Extended Data Fig. 1 and 2) and blood gas analysis (seen in Fig. 5 and Extended Data Table 1), used to directly assess diaphragm displacement and to evaluate the repercussions on gas exchange, respectively. Characterization of the actuators was performed both *in vitro* and *in vivo*. We have expanded and reworked the discussion to incorporate further considerations for clinical translation, including current limitations, clinical concerns, and the future preclinical work needed before human implantation. We feel that these additions to our manuscript have substantially strengthened our claims.

Both reviewers comment on aspects of device characterization. Therefore, we have included two additional figures in the Extended Data (Extended Data Fig. 1 and 2), with a detailed discussion included in the section titled "*Controlling PAM actuator performance via pressurization*" in the Supplementary Notes. Additionally, both reviewers comment on discussing the considerations for translation. Our discussion has been largely rewritten and expanded to address the additional insights provided by the reviewers. Detailed individual responses to each reviewer comment are included below.

Reviewer #2

Overall Reviewer Comment

This paper is the first demonstration of an in vivo implanted system for assisting in breathing and for contracting diaphragm. The use of external ventilators is very critical, and having an implanted system would solve major problems for patients with diaphragm dysfunctions.

I appreciated the paper: the motivations are clear, the in vivo tests are sufficiently solid (even if quite short) and the future impact could be dramatic.

On the other hand, I have identified some major and minor issues which prevent me to give a positive assessment.

Response to Overall Comment

We agree with Reviewer #2's summary of the significance of our work. We believe the reviewer's comments have significantly improved this body of work and hope our responses are to their satisfaction.

Comment 1

First of all, the engineering part is quite limited. All the used components are not optimized for the application. Soft actuators design, size and distribution are not considered in the study protocol (see also comment 4). Sensors and controllers are commercial. The most relevant parts of the paper are the animal test and the physiological measurements.

Response to comment 1

We thank the reviewer for this comment, and we agree that the focus of our paper is not in presenting any individual actuator, sensor, or controller. Instead, we demonstrate the full integration of soft actuators fitted to the diaphragm operated by a custom-built control box that provides device actuation synchronized to native respiratory effort, and ultimately study the interaction and augmentation of this soft robotic system on *in vivo* respiratory physiology. This body of work aims to prove feasibility and proof-of-concept, serving as the first step in the long journey towards translating this work to the clinic.

We fully agree that our initial presentation has obscured our work done to develop the soft robotic system for this application, and we have worked to update the manuscript to better communicate this. Please see updates to the supplemental Information titled "*Controlling PAM actuator performance via pressurization*" and Fig. S1 and S2 for an improved description of the characterization work of the system.

A specific response regarding soft actuator design, size, and distribution can be found in the response to comment 4.

Comment 2

The clinical motivation of this work is very clearly presented at the beginning. The authors are invited to discuss (possibly in the discussion section) if the target patients for this kind of device are eligible to undergo such complex implant surgery procedure. This might be crucial to foresee future clinical employments of the device.

Response to comment 2

Again, we appreciate this comment. Due to the mechanical nature of the device—as opposed to biopharmaceutical interventions—this technology is not restricted to a single disease type, and instead the target patient population is defined by patients with mechanical respiratory failure involving the diaphragm who would benefit from long-term mechanical augmentation of diaphragm function. Although such a surgical intervention is invasive and might lead to complications, it has been well demonstrated that thoracic surgery is feasible in patients with peri-operative risk (e.g. lung transplant for terminal respiratory failure). Therefore implantation of our device may be considered, especially in a highly skilled environment. We therefore added the following paragraphs regarding considerations for the translation to the clinical field to the discussion of the manuscript.

Towards clinical translation

Envisioning a translation to the clinical field, the following considerations might help to optimize the management and pave the way to human application. The diseases leading to chronic diaphragmatic dysfunction are numerous and feature very different pathophysiologies. Therefore, a thorough understanding of the underlying pathology as well as its specificity are critically needed to help optimize the management and anticipate complications⁴⁶. Moreover, patient selection and indication will need to be clearly defined in order to select the patients who will benefit from this therapy the most. Here, we present a generalized mechanical strategy for diaphragm support, but the parameters of actuator design or actuation control will need to be optimized and specialized per the needs of a given pathology as well as individual patient anatomy.

Owing to the complexity of the procedure, a multidisciplinary team highly trained in advanced thoracic surgery is required to build expertise and develop this technology, ideally in a high-volume center⁴⁷. Technological improvement is required to provide the least invasive approach of implantation. In this regard, a thoracoscopic route might be beneficial and will be the subject of future work. Given the invasive nature of implantable devices, the diaphragm assist platform is targeted towards patients with chronic-to-permanent ventilator dependence. We recognize that surgery in patients suffering severe diaphragm dysfunction causing respiratory failure can carry a high morbidity and mortality. Peri-operative complication can be numerous; one of the most feared is the worsening of the pulmonary status, which may itself precipitate the need for long-term ventilation⁴⁸. Nevertheless, it has been well demonstrated that complex thoracic surgery is feasible even in very frail patients. Lung transplantation for terminal respiratory disease⁴⁹ is one of the most striking examples. Thus, surgery could still be considered in a suitable target population that would ultimately benefit from this mechanical augmentation of

diaphragm function, such as a range of neuromuscular disorders. The concept of diaphragm assist is in itself a means of preventing further complications from chronic respiratory failure and preserving key aspects of quality of life—like speech and mobility.

Comment 3

In the introduction section the authors should better contextualize their work not only with reference to soft actuators but also to fully implantable robots. Few examples and field definition were recently reported in the state of the art. Just as examples, please refer to: - Damian, D. D., Price, K., Arabagi, S., Berra, I., Machaidze, Z., Manjila, S., ... & Dupont, P. E. (2018). In vivo tissue regeneration with robotic implants. *Science Robotics*, 3(14). - Iacovacci, V., Tamadon, I., Kauffmann, E. F., Pane, S., Simoni, V., Marziale, L., ... & Menciassi, A. (2021). A fully implantable device for intraperitoneal drug delivery refilled by ingestible capsules. *Science Robotics*, 6(57), eabh3328. - Menciassi, A., & Iacovacci, V. (2020). Implantable biorobotic organs. *APL bioengineering*, 4(4), 040402.

Response to comment 3

We agree that the introduction is better served with a broader and more recent context of fully implantable robots. The text has been updated to:

Previously, fully implanted soft actuators have shown the ability to augment heart function^{7–11} and many other newly developed implantable robotics have shown utility in a broad spread of biological applications^{12–20}.

With the additional citation of the following sources:

9. Payne, C. J. *et al.* An Implantable Extracardiac Soft Robotic Device for the Failing Heart: Mechanical Coupling and Synchronization. *Soft Robot.* **4**, 241–250 (2017).
10. Saeed, M. Y. *et al.* Dynamic Augmentation of Left Ventricle and Mitral Valve Function With an Implantable Soft Robotic Device. *Basic to Transl. Sci.* **5**, 229–242 (2020).
11. Hong, Y. J., Jeong, H., Cho, K. W., Lu, N. & Kim, D. H. Wearable and Implantable Devices for Cardiovascular Healthcare: from Monitoring to Therapy Based on Flexible and Stretchable Electronics. *Adv. Funct. Mater.* **29**, 1808247 (2019).
14. Damian, D. D. *et al.* In vivo tissue regeneration with robotic implants. *Sci. Robot.* **3**, (2018).
15. Iacovacci, V. *et al.* A fully implantable device for intraperitoneal drug delivery refilled by ingestible capsules. *Sci. Robot.* **6**, 3328 (2021).
16. Dolan, E. B. *et al.* An actuatable soft reservoir modulates host foreign body response. *Sci. Robot.* **4**, (2019).
17. Menciassi, A. & Iacovacci, V. Implantable biorobotic organs. *APL Bioeng.* **4**, 1–4 (2020).
18. Perez-Guagnelli, E. *et al.* Characterization, Simulation and Control of a Soft Helical Pneumatic Implantable Robot for Tissue Regeneration. *IEEE Trans. Med. Robot. Bionics* **2**, 94–103 (2020).

19. Pane, S., Mazzocchi, T., Iacovacci, V., Ricotti, L. & Menciassi, A. Smart implantable artificial bladder: An integrated design for organ replacement. *IEEE Trans. Biomed. Eng.* **68**, 2088–2097 (2021).
20. Amiri Moghadam, A. A. *et al.* Using Soft Robotic Technology to Fabricate a Proof-of-Concept Transcatheter Tricuspid Valve Replacement (TTVR) Device. *Adv. Mater. Technol.* **4**, (2019).

Comment 4

It is not clear how the number of PAMs and the overall output force required were set. Please clarify on this and comment on the role played by spatial distribution of the two actuators on the diaphragm surface. What about the needed PAM contraction? Should this be adjusted for each subject and then be constant over the respiratory task?

Response to comment 4

We agree that our initial manuscript did not clearly communicate these aspects of the work.

In brief, shape of the pressurization curve was chosen via a comparison of different input waveforms seen in the data presented in Extended Data Fig. 1 with the following conclusion in the Supplemental Notes: “Overall, the curved waveform used in the majority of the study represents a pressurization scheme that aims to optimize between the tradeoffs of the square wave and triangular wave, generating the best tidal volumes and biomimetic PV loops.”

The pressurization curve was scaled to different depths of pressurization. Because we are aiming for maximal augmentation in the state of simulated diaphragm failure, this study actuates the device at a maximum of 20 psi, captured in Extended Data Fig. 2 and described via the following statement in the Supplemental Notes: “The degree of pressurization has a positive, but nonlinear effect on the amount of diaphragm displacement generated (Extended Data Fig. 2g-k).”

In cases where less augmentation is needed, level of augmentation can be tuned via the pressurization. Please see the new section of the Supplemental Information titled “Controlling PAM actuator performance via pressurization” and Extended Data Fig. 1 and 2 for full details regarding how the overall output force and device performance was determined by controlling the pressurization waveform and depth of pressurization.

The choice to use two actuators—one on each hemidiaphragm—was due to the issue that the introduction of each actuator displaces potential lung volume in the thoracic cavity. The filled actuator size is 17 mL at 20 psi, so the use of additional actuators would require augmentation that provides benefits beyond this filled volume. The two hemidiaphragms are semi-independent (Whitelaw WA. *J Appl Physiol* 62, 180-6 (1985)), meaning that augmentation and displacement on one side doesn't transfer to the other, which is why clinically some people experience unilateral diaphragm paralysis (Gibson G. *Thorax* 44, 960-970 (1989), Celli, B. *Semin Respir*

Crit Care Med 23, 3, 275-281 (2002)). The use of two actuators represents the minimum volume of actuators to act upon both semi-independent hemidiaphragms. For a potential alternative design, actuators that provide greater coverage of the diaphragm with minimal increase in occupied thoracic volume could provide a greater surface area to distribute force and provide more homogeneous diaphragm displacement. Designing actuators that lend themselves to this configuration is one direction of future work that the diaphragm assist system would benefit from.

Regarding the spatial distribution of the two linear PAM actuators used in this study, the PAMs are placed in the antero-posterior direction, the left actuator being lateral to the heart. Specific placement is reflected in the methods text: “The anterior portion is attached to the sternum and the posterior attachment is made to the lowest posterior rib in the most medial position that can be achieved without disrupting the region of the major arteries and veins, esophagus, and spine.” This is best visualized via the coronal view of the fluoroscopy in Supplemental Video 1. This placement aims to place the PAM across the peak of the dome of the actuator and laterally down the rostral and dorsal edge of the subject’s ribcage.

We agree that the PAM contraction necessary to augment diaphragmatic motion is a key design feature. At zero pressurization, the actuator should passively fit along the peak arclength of the end expiratory diaphragm. The theoretical maximum contraction length should be the straight line distance between the two points of attachment. In order to anchor the actuator via sutures, the actuator requires some region of passive material. We developed the reported actuator dimensions empirically in accordance with our work in swine cadavers to appropriately fit the arclength and anatomy of the diaphragm in 30-40kg subjects. The actuator reported here has a relaxed length of 25 cm, and a contracted length of 21cm. There exists approximately 3.5 cm of PET mesh at each end of the actuator which is used as a suture attachment location. We agree that the actuator size and level of contraction can and should be fitted to individual patients. However, given the nature of animal experiments, we do not have access to “patient data” and “patient dimensions” ahead of the procedure. We control for variation in patient dimensions by controlling swine size, using only female Yorkshire swine in the 30-40kg range. The actuator is implanted in a manner where the actuator fits snugly against the diaphragm, with the 7 cm of extra mesh as a buffer for interanimal variability. A next generation actuator for this diaphragm assist system could be designed in a manner that is not limited by the straight line distance and could also provide additional downward motion beyond the geometric “chord” of the arclength of the diaphragm. This consideration is briefly captured in the discussion via: “We present a generalized mechanical strategy for diaphragm support, but the parameters of actuator design or actuation control will need to be optimized and specialized per the needs of a given pathology as well as individual patient anatomy.”

Comment 5

The authors employed a custom controller aimed at synchronizing the soft robotic actuator and natural breathing. However, both sensing (spirometry) and control rely on external (bulky?) data acquisition and control platforms. This appears a bit in contrast with the original motivation to outperform over invasive ventilation systems. No concrete paths towards implantations are given.

Response to comment 5

We agree that the current control system and synchronization method prevent its application in an outpatient setting, which represents the long-term objective of this type of device. The present work was designed as a proof-of-concept study. For our purposes, we developed a control system whose components can be easily accessed, modified, and then optimized. Consequently, the size of the current system prevents the outpatient use.

We therefore added the following section in the discussion to present concrete paths toward this endeavor:

An ideal next generation control system aims to trigger from a more upstream neural signal—such as the electrical activity of the diaphragm—to provide an earlier signal that enables an advanced control system to optimize synchronization, removing delays and asynchrony. Neural triggering via implanted electrodes would also untether the current system from the flow instrumentation, freeing the patient from interventions at the mouth or trachea. To fully realize untethering from bulky machines—like standard mechanical ventilators—the external components that control and power the system require miniaturization. Future work will aim to eventually miniaturize the system to the scale of a small backpack—one that could be worn by the patient or attached to an electric wheelchair. The process of miniaturization and portability has proved to be possible in similar complex devices, such as ventricular assist devices (e.g. Thoratec HearMate III) or total artificial hearts (e.g. Syncardia TAH, Carmat Aeson)^{42–45}.

In combination with the following text from the original manuscript:

We envision further translational potential of this technology when combined with the development of smaller and more portable pneumatic energy sources^{54,55} as the field of soft robotics advances. With the integration of a portable pump and control system in the future, this technology could provide an additional level of patient autonomy via increased mobility.

Comment 6

The caption of figure 2 is very informative. However, it would be useful to add some labels on the axes, graphs title, etc., to make figures readability more straightforward.

Response to comment 6

We thank the reviewer for pointing this out. We have added more labels and graphics to the original Figure 2 (now Figure 3) to improve readability. Specifically, we have added gray dashed lines to visually indicate the selection of data in Fig. 3c-e from Fig. 3b. and labeled the green "Range of Normal"

Comment 7

The *in vivo* test is very short, so all considerations about materials, biocompatibility, stability etc. are missing.

Response to comment 7

We agree that the acute nature of the study did not allow for important longer term testing of materials and stability.

Regarding stability, we conducted acute fatigue testing, showing resilience of the actuators to at least 3000 cycles of actuation *in vivo* without failure at the end of the study. Actuators for this study were designed to last through a one-day sacrificial study. Future work necessitates full characterization of the lifetime of updated actuators. This will be a key future design requirement.

Regarding materials and biocompatibility, the actuators were manufactured from commercially available materials for ease of prototyping. We did consider the future biocompatibility of the device during material selection, opting for materials with established, regulatory-approved counterparts. We have updated the discussion to address this limitation as follows:

Due to the focus on feasibility, we acknowledge that there are limitations in these acute studies from the lens of regulatory approval and clinical translation. We do not study device biocompatibility or long-term device operation. The device was constructed from types of polymers that are already used in established medical devices⁵⁰⁻⁵³, such as poly(ethylene terephthalate) (PET) and polyurethanes (See Supplemental Information). Because our device focuses on mechanical interaction, as opposed to biochemical interactions with the body, the materials used in the device can easily be substituted with regulatory-approved materials in future iterations. With improved performance and stability, future long-term studies will need to investigate the long-term effects of the system including tissue remodeling and the ability to provide full-time respiratory support.

Further details on material selection can be found in the updated Supplemental Information section titled *Considerations for PAM materials selection*.

Comment 8

In line 512, 6 pigs are mentioned. But in the rest of the paper we had 5 pigs.

Response to comment 8

We thank the reviewer for pointing out the need for clarification. In our first submission, the original Fig. 3 displays the data for the 6 total pigs mentioned in line 512 (named A-F) when testing synchronized vs. independent actuation. The original Fig. 2 does in fact only display the data for 5 total pigs (A-E) because for pig F, we did not collect a sufficient amount of the baseline data during a period of unsupported ventilation to make the comparison shown in the original Fig. 2.

For this revised submission, given the additional data gathered and used for the updated Figures 2, 4, 5, S1, and S2, we have updated the referenced text accordingly: “We used a total of **twelve** swine during the development and testing of our system, and we present data from **nine** swine in the manuscript. **Different subsets of subjects were used for the experimental investigations reported; not all subjects were used in every experimental investigation.**”

Reviewer #3:

Overall Reviewer Comment

Overall This work by Drs. Hu and Roche developed an implantable device that can substitute a mechanical ventilator commonly used in contemporary clinical care and yet, avoid the various shortcomings of it. The authors build a robotic actuator which is overlaid to the subject's diaphragm and activated by the host's inspiratory effort, which in turn, is balloon inflated and substitutes the function of the native diaphragm.

The authors challenge an important clinical question, with a good understanding of the respiratory cycle and the role of the diaphragmatic motion. The concept underlying the current work is interesting and innovative. I acknowledge that this is a prototype, however the current manuscript do have room for improvement.

Additionally, considering the invasiveness of the implantation associated with their device, whether the strength of the system is enough to overcome the invasiveness associated with the procedure should be discussed more in depth.

Overall Response Response

We agree with Reviewer #3's summary of the significance of our work and the proof-of-concept status of this study. We believe the reviewer's comments have significantly improved this body of work and hope our responses are to their satisfaction.

Additionally, we thank the reviewer for this relevant comment specifically on the considerations between performance and invasiveness. We demonstrated in this work the ability of the device to significantly augment respiratory function, such as tidal volume and minute ventilation, and the ability to restore a normal range of minute ventilation. We further investigated *in vivo* the effect on gas exchange using blood gas analysis (see *Response to comment 1*) and explored diaphragm motion using ultrasound. The performance of the device demonstrates strong feasibility of this soft robotic strategy, but we acknowledge that there are limitations and “A core goal of the next generation system is to further improve the tidal volume augmentation, which will need to be achieved through both actuator design and control system development.”

Establishing more robust performance and longer respiratory trials will be necessary for future translation.

In addition, updates to the external triggering and control systems will be critical for realizing our vision of untethering from bulky machinery. This future work is proposed in our discussion section: “Neural triggering via implanted electrodes would also untether the current system from the flow instrumentation, freeing the patient from interventions at the mouth or trachea. To fully realize untethering from bulky machines—like standard mechanical ventilators—the external components that control and power the system require miniaturization. Future work will aim to eventually miniaturize the system to the scale of a small backpack—one that could be worn by the patient or attached to a belt or an electric wheelchair.” Given an improved next-generation system that can provide long-term ventilatory support to an appropriate patient population, the

rescue of respiratory performance that avoids further worsening of patient status and preserves key quality of life measures—like speech and mobility—may justify the surgical procedure.

We agree that such a surgical intervention is invasive and may lead to complications. However, it has been well demonstrated that thoracic surgery is feasible in patients with high peri-operative risk (e.g. lung transplant for terminal respiratory failure). Therefore implantation of our device may be considered, especially in a highly skilled environment. We therefore add the following paragraphs in the discussion, and make general considerations for the translation to the clinical field.

Towards clinical translation

Envisioning a translation to the clinical field, the following considerations might help to optimize the management and pave the way to human application. The diseases leading to chronic diaphragmatic dysfunction are numerous and feature very different pathophysiologies. Therefore, a thorough understanding of the underlying pathology as well as its specificity are critically needed to help optimize the management and anticipate complications⁴⁶. Moreover, patient selection and indication will need to be clearly defined in order to select the patients who will benefit from this therapy the most. Here, we present a generalized mechanical strategy for diaphragm support, but the parameters of actuator design or actuation control will need to be optimized and specialized per the needs of a given pathology as well as individual patient anatomy.

Owing to the complexity of the procedure, a multidisciplinary team highly trained in advanced thoracic surgery is required to build expertise and develop this technology, ideally in a high-volume center⁴⁷. Technological improvement is required to provide the least invasive approach of implantation. In this regard, a thoracoscopic route might be beneficial and will be the subject of future work. Given the invasive nature of implantable devices, the diaphragm assist platform is targeted towards patients with chronic-to-permanent ventilator dependence. We recognize that surgery in patients suffering severe diaphragm dysfunction causing respiratory failure can carry a high morbidity and mortality. Peri-operative complication can be numerous; one of the most feared is the worsening of the pulmonary status, which may itself precipitate the need for long-term ventilation⁴⁸. Nevertheless, it has been well demonstrated that complex thoracic surgery is feasible even in very frail patients. Lung transplantation for terminal respiratory disease⁴⁹ is one of the most striking examples. Thus, surgery could still be considered in a suitable target population that would ultimately benefit from this mechanical augmentation of diaphragm function, such as a range of neuromuscular disorders. The concept of diaphragm assist is in itself a means of preventing further complications from chronic respiratory failure and preserving key aspects of quality of life—like speech and mobility.

Major comments

Here are some major comments for the authors.

Comment 1

The authors fail to analyze the actual blood gas levels with arterial blood. The authors state that the responsiveness of the diaphragm can vary between subjects. How can the authors be sure that the blood gas levels are maintained at a relatively stable state with the device instillation? Please be aware that maintaining the blood acid-base balance in a homeostatic state is a critical role of respiration in every living creature.

Response to comment 1

We agree that arterial blood gas (ABG) levels are the clinical gold standard indicator of respiratory status.

To this end, we have included an additional section, figure (Fig. 5), and Extended Data Table 1, capturing the effect of synchronization on the blood gas balance. The manuscript text and figure are copied below:

Effect of synchronization on blood gas exchange

Physiologically, ventilation is necessary to bring in oxygen (O_2) and to clear out accumulated carbon dioxide (CO_2) from the blood. Arterial blood gases (ABGs) are discrete blood analyses that give a snapshot view of the gas exchange and acid-base homeostasis, providing measurement of partial pressure of O_2 (P_aO_2) and CO_2 (P_aCO_2), pH, and bicarbonates (HCO_3^-) in arterial blood. P_aCO_2 is directly and inversely proportional to alveolar ventilation and is therefore a representative metric of ventilatory function. Only pH and pCO_2 are depicted here in Fig. 5, but the full ABG parameters are reported in Extended Data Table 1 and discussed in the Supplementary Notes.

As shown in the prior section, the high variance from independently actuated ventilation showed mixed constructive and destructive interference (Fig. 4e,f) which led to worse ventilation outcomes. The same variance in the peak inspiratory flows and tidal volumes over time due to independent vs. synchronized actuation can be seen in Fig. 5a and Fig. 5b. In these two respiratory challenges, the subject was switched directly from the standard mechanical ventilation to our diaphragm assist system, evaluating its ability to maintain gas exchange.

In the respiratory challenge operated with independent actuation (Fig. 5a), we see high levels of hypercarbia over time. As a result, respiratory acidosis develops, which is a direct consequence of increased P_aCO_2 (Extended Data Table 1a). Contrastingly, in a respiratory challenge operated with synchronized actuation in the same animal (Fig. 5b), pCO_2 levels are relatively well maintained. The acidemia observed for this trial is rather of metabolic cause (called metabolic acidosis) (Extended Data Table 1b, see Supplementary Notes).

In another experiment, a respiratory trial was initiated with 2 minutes of unsupported ventilation and then switched to our diaphragm assist system, evaluating its ability to recover from a period of unsupported ventilation. During the 2 minutes of unsupported ventilation, high levels of CO₂ accumulate quickly over this brief amount of time (Fig. 5c). After two minutes, the diaphragm assist system is actuated with synchronized actuation. The increasing acidification and accumulation of CO₂ reverses and some recovery from the hypercarbic state is seen in the first 10 minutes, with a slight uptick in the CO₂ around 15 minutes into the challenge.

Fig. 5: ABGs taken across distinct respiratory challenges. **a**, In a respiratory challenge operated with independent actuation, a representative set of peak actuation pressure, peak inspiratory flow, and tidal volumes, and the pH and pCO₂ values from discrete arterial blood gases taken. **b**, In a respiratory challenge operated with synchronized actuation, a representative set of peak actuation pressure, peak inspiratory flow, and tidal volumes, and the pH and pCO₂ values from discrete arterial blood gases taken during one full respiratory challenge with synchronized actuation. The respiratory challenges depicted in **a**, and **b**, are taken from the same animal. **c**, In another animal, a respiratory challenge began with a 2 minute period of unsupported ventilation and subsequent synchronized actuation. A representative set of peak actuation pressure, peak inspiratory flow, and tidal volumes, and the pH and pCO₂ values from discrete arterial blood gases taken. Gray shading indicates the period of time where the system is off and respiration is unassisted. Light green shading indicates the standard range

of normal values for each arterial blood gas metric. Complete ABGs can be found in Extended Data Table 1.

Due to their discrete nature, in contrast to the continuous signals like flow and volume collected by our data acquisition system, the ABG data is much sparser because ABGs were only taken during a subset of respiratory challenges. For the respiratory challenges where ABGs were taken, we collect them every 2 or 5 minutes (2 minutes to capture the effect of 2 minutes of unsupported ventilation, as seen in Fig. 5c). This is captured in the Methods section with the following text:

For experiments investigating gas exchange, ABGs were collected at 2 or 5-minute intervals during the challenge.

Comment 2

How is the level of inspiration controlled with their device? The degree of inspiration should be controllable according to the subjects needs in various clinical situations (for example, in early sepsis, it is natural to be in a hyperventilated state) but the authors do not show that their device is versatile enough to accommodate this. What happens to the subject's respiration if the bladder in the system is pressurized more? Additionally, what about the opposite where there is no need for a deep inspiration?

Response to comment 2

We thank the reviewer for this relevant comment. We agree that in a clinical setting, the degree of inspiration should be tunable. To demonstrate this capacity, we conducted significant further experimentation that substantially improved our understanding of our device.

The degree of assistance provided by the device is mainly determined by the level of pressurization of the PAM, which is set by the user in the control system. A pressure actuation curve (selected by the user), once triggered, is transmitted to the electropneumatic regulator. In the present study, we simulate severe diaphragmatic failure in all subjects, reflected by the low unassisted tidal volumes (see Fig. 3). Therefore, the maximal possible augmentation was required to maintain respiratory homeostasis for all the subjects in this study.

Looking towards future translation, as pointed out by the reviewer, the subject might need different degrees of assistance (i.e., different levels of pressurization), depending on the clinical situation and metabolic state. To further characterize the device and understand the effect of different pressurization of the PAM on respiratory mechanics, we conducted a series of *in vitro* and *in vivo* experiments during which we adjusted the degree of pressurization (5, 10, 15, and 20 PSI). The characterization data regarding degree of pressurization is presented in our new Extended Data Fig. 2.

Extended Data Fig. 2. Tuning actuation depth via level of pressurization. The actuator pressure profile for a curved waveform scaled to have a peak nominal pressure of (a) 5 psi, (b) 10 psi, (c) 15 psi, (d) 20 psi. The peak forced generated by different levels of actuation were characterized *in vitro* on a (e) classic Instron tensile test setup and (f) our modified flexural test setup (depicted in Fig. S5). Diaphragm displacement generated by actuations of (g) 5 psi, (h) 10 psi, (i) 15 psi, (j) 20 psi visualized via M-mode ultrasound. (k) The average diaphragm displacement per breath from one sample subject via M-mode ultrasound. (l) Tidal volume achieved via different levels of pressurization from one sample subject. Significance is indicated by * $p < 0.05$ for a two-sample t test. (m) Respiratory Campbell diagram plotting the pleural pressure-volume loops for representative breaths from different levels of actuation.

This data supports the following update to the text of the main manuscript:

Actuator behavior is governed by the degree of pressurization. Set pressurization waveforms are programmed to the control system and electropneumatic regulators. *In vitro* and *in vivo* characterization of actuator behavior when controlled by different pressurization waveforms is included in the Extended Data (Extended Data Fig. 1 and Extended Data Fig. 2).

Details for this characterization are updated in the Methods section of the manuscript as follows:

Actuator characterization was conducted both *in vitro* and *in vivo*. For the *in vitro* characterization, actuator performance was measured via Instron testing. Classic tensile testing was conducted to measure the contractile force. A modified flexural bend setup (Fig. S5) was used to measure the perpendicular force applied to the diaphragm via arclength shortening. For the *in vivo* characterization, performance of the diaphragm assist system was evaluated through the diaphragm displacement (via ultrasonography) and the functional metrics (tidal volume, Campbell diagram) (Extended Data Fig. 1 and 2). Different pressurization shapes and levels were input into the actuator (Extended Data Fig. 1 and 2) and the resulting behavior was measured. Further details can be found in the Supplementary Information.

Detailed discussion of this new characterization data regarding different levels of pressurization are found in the below portion of the *Controlling PAM actuator performance via pressurization* section of the Supplemental Notes:

PAM performance can also be tuned via depth of pressurization by scaling the input curved waveform shown in Extended Data Fig. 2a to different peak pressures (5, 10, 15, and 20 psi), the resulting actuator pressure waveforms are shown in Extended Data Fig. 2a-d. The relationship between pressurization and forces generated is linear (Extended Data Fig. 2e,f) which corroborates previous McKibben characterization work²⁴. We characterize the response of one subject to the varying degrees of pressurization. The degree of pressurization has a positive, but nonlinear effect on the amount of diaphragm displacement generated (Extended Data Fig. 2g-k). In this subject, we demonstrate tunability of the degree of augmentation via changes in pressurization, with the greatest range of responsiveness being between 0 and 10 psi. Additional increases taper off between 10 and 20 psi, which matches the understanding of how McKibben actuators operate, as they first expand and fill to their maximum volume, achieving maximum contraction, and beyond that they increase force generation²⁴. In terms of respiratory mechanics, the degree of pressurization does not have a large effect on the change in pleural pressure (Extended Data Fig. 2m) unlike the different waveform shapes in Extended Data Fig. 2p.

Notably, interanimal variability is undeniably a factor contributing to overall performance, as evident in the varied responsiveness to the device seen in Fig. 3. Even in a case of low augmentation, we saw a nonlinear but tunable response to different levels of pressurization. The absolute degrees of augmentation shown in Extended Data Fig. 2k-m will obviously not hold across different animals, but we expect that the relative effect of tuning pressure should.

Comment 3

Likewise, I would like to see how the respiratory mechanics change according to how fast the bladder in their system is inflated. More information should be given on this. The respiratory mechanics are shown as simple bar graphs in Figures 5a, b and c, whereas the gradient or slope of the Campbell diagram may also be important.

Response to comment 3

We have included a robust characterization of the effect of changing the manner of bladder inflation in the Supplemental Notes section titled *Controlling PAM actuator performance via pressurization* and in Extended data Fig 1 reproduced below, demonstrating the effect of different input waveforms on actuator forces and *in vivo* respiratory mechanics.

Extended Data Fig. 1. Controlling actuation via different pneumatic waveforms. Input waveforms of a (a) curved, (b) square, and (c) triangle shape can be programmed into the

custom-built control system. The effective output pressure of the electropneumatic regulator for the (d) curved, (e) square, and (f) triangle shape drives actuation. The PAM actuation forces were characterized for different waveforms *in vitro* on a classic Instron tensile test setup (g,h,i) and our modified flexural test setup (j,k,l) (depicted in Fig. S5). Input waveforms of a (m) curved, (n) square, and (o) triangle shape generate different shapes of diaphragm displacement as visualized via M-mode ultrasound. (p) Average diaphragm displacement from m,n,o. (q) Average tidal volume and (r) respiratory Campbell diagram plotting the pleural pressure-volume loops for representative breaths from different waveform shapes. Significance is indicated by * $p < 0.05$, ** $p < 0.01$, *** $p < 0.001$ for a two-sample t test.

Detailed discussion of this new characterization data regarding rates of pressurization are found in the below portion of the *Controlling PAM actuator performance via pressurization* section of the Supplementary Notes:

Different input shapes explore the effect of rate of pressurization (Extended Data Fig. 1a-c). The fidelity to these idealized waveforms is limited by the control resolution of the electropneumatic regulators, and ultimately result in the output pressurization curves of Extended Data Fig. 1d-f. These actuation pressure curves ultimately govern the mechanical performance of the actuators. The actuators are characterized *in vitro* via tensile and flexural testing, as described in the Supplemental methods. The tensile force (Extended Data Fig. 1g-i) represents the contractile force applied to the points of attachment on the ribs, and the flexural force (Extended Data Fig. 1j-l) represents the force perpendicular to the actuator towards the diaphragm.

Different actuation pressure waveforms result in different displacements, (seen in the M-mode ultrasound in Extended Data Fig. 1m-o and quantified in Extended Data Fig. 1p), tidal volumes (Extended Data Fig. 1q) and different respiratory mechanics (Extended Data Fig. 1r). Notably, the square wave pressurization is distinct from the behavior of the curved wave and triangle wave, especially with regards to the average diaphragm displacement and the Campbell diagram. We note that the square wave achieves similar tidal volumes to the other waveforms while drawing more negative pleural pressures. The slope of the Campbell diagram, taken at the two points in the loop where , can be viewed as a representation of compliance of the system. A negative pleural pressure drives flow via the gradient from atmospheric pressure at the airway opening to the negative alveolar pressure, so we evaluate the absolute value of compliance. The slope generated by the square wave (12.1 mL/cmH₂O) is considerably lower (i.e., the system is stiffer) than that of the curved (15.9 mL/cmH₂O) and triangle wave (15.1 mL/cmH₂O), which both have slopes that more closely resemble those that of spontaneous respiration (25.6 mL/cmH₂O). These values are overall relatively stiff and are likely due to the low lung volumes for this subject. Qualitatively, we observe that the square wave results in “sharper” breaths that pull on the chest wall more aggressively compared to the gentler inflation of the curved and triangle wave, matching the much higher tensile forces generated by the square wave from *in vitro* testing.

The curved and triangular pressurization input are similar in their pressure-volume (PV) loops; however, the curved input achieves higher tidal volumes with marginally smaller levels of

diaphragm displacement (Extended Data Fig. 1p-r), which could be attributed to the shorter time in which the actuator operates at a high pressure filled state which may not provide enough time for lung filling. Overall, the curved waveform used in the majority of the study (Fig.2-7 in the main text) represents a pressurization scheme that aims to combine the benefits of the square wave and triangular wave, generating the best tidal volumes and biomimetic PV loops.

Methods for this characterization in Extended Data Fig. 1 are described along with the methods of Extended Data Fig. 2 in Supplementary methods *McKibben PAM Mechanical Characterization Methods*.

Comment 4

The authors should be commended for doing a nice job in synchronizing the device to the subject's respiratory effort. However, the trigger for this is the spirometer that is installed into the subject's airway. To really overcome the weakness of the contemporary mechanical ventilators, the authors should think of better methods that do not touch the respiratory tract to sense the intrinsic respiratory effort.

Response to comment 4

We agree that a synchronization method that does not involve the patient airways is required to overcome the tethered limitations of contemporary ventilators and will be required for clinical translation. We therefore added two additional sections in the discussion to discuss this limitation and provide pathways for future work as follows:

First,

Some neuromuscular signals, like the electrical activity of the diaphragm (Edi), contain detailed information about both inspiration and expiration times^{35,36}. Edi amplitude is also proportional to the neural drive, as well as the degree of contraction of the diaphragmatic muscle, therefore opening up the possibility of adaptive control. Triggering from Edi measured at the esophageal level via a feeding tube³⁷ may be warranted to improve mechanical ventilation. This method, known as neurally adjusted ventilatory assist, is available in the clinical setting with mechanical ventilation and may improve respiratory weaning of patients that are challenging to wean³⁶. The same principle could be applied to our diaphragm assist system; using a more upstream signal with greater information on the native respiratory effort would allow for a more robust control system.

Second,

Synchronization is critical to device performance, and thus future work lies in building a next generation control system; this includes creating a system that is cognizant of the beginning of expiration as opposed to inspiration, an automated control system that removes the error of manual titration, and further investigation of dynamic actuation curves. An ideal next generation control system aims to trigger from a more upstream neural signal—such as the electrical activity

of the diaphragm—to provide an earlier signal that enables an advanced control system to optimize synchronization, removing delays and asynchrony. Neural triggering via implanted electrodes would also untether the current system from the flow instrumentation, freeing the patient from interventions at the mouth or trachea.

Minor comments

There are also some minor comments that might help.

Comment M1

Supplementary Video Online is not that informative as it plays in a real-time. It would be better if the video is played more slowly.

Response to comment M1

We have amended the video to play both in real-time and at a 0.3x speed for clarity.

Comment M2

The finding should be supported by adequate statistical analysis.

Response to comment M2

We appreciate this comment, and have detailed the statistical analysis presented in the section “Statistical analysis” in the Methods included here. We have also responded to the specific reviewers comments in the subsequent subcomments.

Statistical analysis

Statistical tests were conducted as described in the respective figure captions for Fig. 3, 4, 7, and Extended Data Fig. 1 and 2. For Fig. 3c,d and Fig. 7a-c, two-sided Wilcoxon rank-sum analyses were conducted in MATLAB (MathWorks, Portola Valley, CA, USA) via the “ranksum” function. Fig. 4e,f depicts two sets of statistical tests. A two-sided Welch’s t-test without assuming equal variances was conducted in order to compare the means of the populations via the “ttest2” function in MATLAB with an “unequal” variance type specification. Additionally, a 2-sample F-test for equal variances was conducted to compare and confirm unequal variances via the “vartest2” function in MATLAB. For the Extended Data Fig. 1 and 2, two-sided t-tests were conducted via the “ttest2” function in MATLAB.

Comment M2.1. Although somewhat obvious from the graphs, there should be some statistical analysis to show the difference in the variation of peak inspiratory flow and tidal volume of Figure 3e and f.

Response to comment M2.1

We have added significance bars for two sets of statistical tests conducted. The black bars represent the significance from a Welch’s t-test to compare the mean of the two populations, and the gray bars represent significance from a 2-sample F-test for equal variances to compare

the variance of the two populations. This is seen in the updated figure caption (Black significance bars are results from Welch's t-test comparing means. Gray significance bars are results from a 2-sample F-test for equal variances comparing variances. Significance is indicated by * $p < 0.05$, ** $p < 0.01$, *** $p < 0.001$ for both statistical tests.) and figure for Fig. 4.

Comment M2.2

The pairwise comparisons should be done for Figure 5a, b, and c with the data acquired at the spontaneous respiration (SR) as the reference, not the actuator assisted ventilation (AAV).

Response to comment M2.2

We have added the additional set of significance bars using SR (comparing SR and mechanical ventilation (MV)) as the reference in Fig. 7a-c.

Comment M2.3

How many samples were taken for each subjects in each graph. This should be noted.

Response to comment M2.3

We have included the range of sample size for each type of plot in the figure captions for the updated Fig. 3c,d ($n = 11-27$ breaths), Fig. 4e,f ($n = 119-419$ breaths) and 7a-c ($n = 11-32$ breaths) in addition to the individual data points on the figures.

Comment M3

In Figures 2c, d and e, when did the authors gather data after the device was turned off or on? The same questions goes for synchronized actuation in Figures 3e and f, Figures 5a, b and c.

Response to comment M3

We have made updates to the figures and figure captions to clarify the data analysis.

The data used in the updated Fig. 3c,d,e (original Fig. 2c,d,e) are gathered from the 30 seconds immediately before and after the device is turned on/off. We have added the dashed gray line graphics to Figure 3 to illustrate this, and have made the following update to the figure caption:

c,d, Comparison of the average (**c**) peak inspiratory flow and (**d**) tidal volume in the 30 second period immediately before and after the point where the assist is turned on at the beginning (left two bars per subject) and off at end (right two bars per subject) of the respiratory challenge (as represented by the arrows in **b** and the gray dashed lines in **b-e**) across 5 subjects ($n = 11-27$ breaths). Each gray dot represents one breath. **e**, Body weight normalized minute ventilation achieved during the 30 second period immediately before and after the assist is turned on at the beginning and off at the end of the respiratory challenge.

The data used in updated Fig. 4e,f (original Fig. 3e,f) span from 300s from the start of the challenge to the end of the respiratory challenge. The 300s exclusion aims to remove the period of time in which the animal's respiratory state adjusts to the new ventilatory state, so that Fig. 4 captures a steady state variance. We have made the following update to the figure caption:

e,f, A swarm plot comparing the steady state (e) tidal volumes and (f) peak inspiratory flows generated with independent actuation and with synchronized actuation for 6 different subjects ($n = 119 - 419$ breaths).

The data used in updated Fig. 7a,b,c (original Fig. 5a,b,c) are gathered from representative 30 or 60 second segments from the data during and surrounding one respiratory challenge per subject. Because spontaneous respiration (SR) and the synchronized actuator assisted ventilation (AAV) both rely on the animal's native respiratory rate (range: 26-32) while the mechanical ventilation (MV) has a set slow respiratory rate (range: 15-20 bpm), 30 second increments were chosen for the SR and AAV segments while 60 seconds increments were chosen for the the MV segments to capture a sizeable sample of breaths. Data segments attempted to capture the closest to steady state achieved in the span of the respiratory challenge. The MV data segments were chosen from the steady state data collected immediately before the initiation of the respiratory system. SR and AAV were chosen from the end of the respiratory challenge, similar to the data selection from the end of the challenge used for updated Fig. 3c,d,e. We have made the following update to the figure caption for updated Fig. 7a,b,c:

a,b,c, Average change in (a) pleural pressure (P_{pl}), (b) abdominal pressure (P_{ab}), and (c) transdiaphragmatic pressure (P_{di}) per breath under mechanical ventilation (MV), actuator assisted ventilation (AAV), and spontaneous respiration (SR) taken from a representative steady-state segment from one respiratory challenge per subject ($n = 11-32$ breaths).

Comment M4

Figure 4a and b miss the legends for y-axis.

Response to comment M4

Thank you. We have added the y-axis labels.

Comment M5

The overall structure of the Discussion could be improved. The contents of each paragraphs tend to jump, making it difficult to follow.

Response to comment M5

We thank the reviewer for this comment and have significantly re-ordered the discussion section. We have also added subheadings (*Contributions, Overall limitations, Towards clinical*

translation) to aid clarity. We hope that it is more readable and flows better after these modifications. The full updated discussion is included here:

Discussion

In this work, we use pneumatic soft robotic actuators to support and augment respiration, demonstrating acute augmentation of physiological metrics of respiration, and feasibility as a proof-of-concept device. A set of two McKibben-style PAMs surgically implanted superior to the diaphragm are capable of providing mechanical support to the diaphragm in a large animal model of respiratory insufficiency. We thoroughly characterized the in vitro mechanical properties of the device and investigated its interactions with the respiratory system and the subject, using multimodal metrics to evaluate respiratory function (e.g. tidal volume, inspiratory flow), biomechanics (cavity pressures, WOB), motion (ultrasonography and fluoroscopy), and gas exchange (ABGs).

Contributions

The diaphragm assist system generated substantial augmentation in respiratory function—measured via peak inspiratory flow (a direct metric of inspiratory function), and tidal volume and minute ventilation (metrics of ventilation)—in our most responsive subject. Subject A had the highest change in peak inspiratory pressure, tidal volume, and minute ventilation; the corresponding large augmentation in peak inspiratory pressure indicates that the volume and minute ventilation augmentation are specifically due to the soft robotic actuators augmenting the diaphragm’s inspiratory function. Responsiveness to the system varied across subjects.

Variance in responsiveness is likely dependent on a combination of many factors. One factor is the level of preserved respiratory baseline. The weak response in the subject with a relatively high preserved weight-normalized minute ventilation (subject E) suggests that the assist system may have weak augmentation or even a disruptive effect in cases of well-preserved diaphragm function. Other potential factors include precise actuator placement, actuator fit, and anatomical variations.

We showed that synchronization with the native respiratory effort is a critical design element in our system. Synchronous actuation is key to consistent, low-variance respiratory waveforms and tidal volumes. Like standard mechanical ventilation, off-cycle actuation of the actuators can lead to a destructive interference with the underlying respiratory effort, resulting in a poor augmentation and poor blood acid-base balance. In evaluating the effect of synchronization on the system’s ability to maintain appropriate gas exchange, we demonstrated that despite generating a similar range of tidal volumes, independent actuation led to an inability to maintain appropriate pCO₂ levels and resulted in respiratory acidosis. Contrastingly, in two trials of well-synchronized actuation, we observed some capacity of the device to maintain and recover baseline pCO₂ levels.

The control system used in this study was a simple but effective first-generation system with many directions for improvement. The synchronization triggered from airway flow—which is also

the metric used by gold standard clinical ventilatory support options for triggering—but flow is also the most downstream signal in neuro-ventilatory coupling. The downstream nature of the signal is a potential source of delays and asynchrony³⁴. In order to achieve consistent assistance from breath to breath, the synchronization must be optimized for the alignment that maximizes constructive interference. The system relied on a manually titrated threshold set for the flow sensor data. It is designed to be triggered at the start of an inspiratory flow effort, which is related to V_0 . However, the manual nature of the system meant that if the threshold was set too low, noise in the flow signal could cause pre-emptive or false triggering (as evidenced by the negative values for P_0-V_0). Our alignment analysis reveals two important considerations for improvements towards this goal. The first consideration is that the influence of alignment changes with the degree of preserved respiratory function, as seen with the difference in results between the intact and the severed phrenic nerve. When the phrenic nerve is severed, all diaphragm motion is governed by the actuators, and misaligned actuation with the remaining native respiratory effort—expansion of the ribcage—results in more consequential destructive interference. Whereas when the phrenic nerve is intact, the net diaphragm motion results from a combination of native diaphragm function and the effect of the actuators, because the actuators only operate along 2 discrete lines on the diaphragm. The contraction of the rest of the native diaphragm motion is still synchronized with the ribcage motion, so the effects of misalignment are less apparent. This implies that optimal alignment parameters may be different for different disease states and the control system will need to be dynamic and adaptive to changes in respiratory function, even within the same patient. The second consideration is that the actuation curve's relationship to the beginning of expiration (V_{pk}) is more influential than the relationship to the beginning of inspiration (V_0). This implies that an updated system should trigger from a signal related to expiration as opposed to the beginning of inspiration. Some neuromuscular signals, like the electrical activity of the diaphragm (Edi), contain detailed information about both inspiration and expiration times^{35,36}. Edi amplitude is also proportional to the neural drive, as well as the degree of contraction of the diaphragmatic muscle, therefore opening up the possibility of adaptive control. Triggering from Edi measured at the esophageal level via a feeding tube³⁷ may be warranted to improve mechanical ventilation. This method, known as neurally adjusted ventilatory assist, is available in the clinical setting with mechanical ventilation and may improve respiratory weaning of patients that are challenging to wean³⁶. The same principle could be applied to our diaphragm assist system; using a more upstream signal with greater information on the native respiratory effort would allow for a more robust control system.

Overall, we show that the strategy to augment the native function of the diaphragm with soft robotics acts as a form of negative pressure ventilation by driving ventilation through the generation of a negative pressure in the thoracic cavity. Our diaphragm assist system is biomechanically similar to that of spontaneous breathing, sharing a substantial portion of the work of breathing in our best responding subject. By functioning as an assist device—as opposed to completely overtaking breathing—our system has the potential to be compatible with voluntary use of the diaphragm. Maneuvers such as voluntary deep breaths or drinking through a straw—abilities related to patient autonomy and quality of life—can be preserved with this implantable ventilator strategy. Additionally, in contrast to current modes of mechanical

ventilation, recapitulation of native biomechanics, as shown with this system, can avoid the deleterious effects that arise secondary to the use of positive pressure ventilation, such as barotrauma^{38,39} or hemodynamic changes in patients with concurrent cardiac pathologies^{40,41}.

Overall limitations

In this study, we demonstrate the foundational work towards a soft robotic implantable ventilator. Translationally, there are many hurdles to overcome between the proof-of-concept state presented here and the ultimately envisioned system, and we discuss them in the subsequent text.

Given that we saw variable responsiveness to the device across subjects, additional studies are needed to understand what factors in system design and implantation can replicate high responsiveness. Our system could generate the low end of acceptable minute ventilations but relied on high respiratory rates to do so. Given the presence of dead space, low tidal volumes result in less alveolar ventilation than if the same minute ventilation achieved with higher tidal volumes and a lower respiratory rate. A core goal of the next generation system is to further improve the tidal volume augmentation, which will need to be achieved through both actuator design and control system development.

Here, we used the classic McKibben actuator; a more application-specific or customized actuator type may allow for further increases in tidal volumes in future work. Other factors in actuator design, such as the number, layout, and positioning of actuators, will also be critical. We demonstrated tunability of assist by controlling pressurization, but an updated design will require finer characterization. Synchronization is critical to device performance, and thus future work lies in building a next generation control system; this includes creating a system that is cognizant of the beginning of expiration as opposed to inspiration, an automated control system that removes the error of manual titration, and further investigation of dynamic actuation curves. An ideal next generation control system aims to trigger from a more upstream neural signal—such as the electrical activity of the diaphragm—to provide an earlier signal that enables an advanced control system to optimize synchronization, removing delays and asynchrony. Neural triggering via implanted electrodes would also untether the current system from the flow instrumentation, freeing the patient from interventions at the mouth or trachea. To fully realize untethering from bulky machines—like standard mechanical ventilators—the external components that control and power the system require miniaturization. Future work will aim to eventually miniaturize the system to the scale of a small backpack—one that could be worn by the patient or attached to a belt or an electric wheelchair. The process of miniaturization and portability has proved to be possible in similar complex devices, such as ventricular assist devices (e.g. Thoratec HeartMate III) or total artificial hearts (e.g. Syncardia TAH, Carmat Aeson)^{42–45}.

Towards clinical translation

Envisioning a translation to the clinical field, the following considerations might help to optimize the management and pave the way to human application. The diseases leading to chronic

diaphragmatic dysfunction are numerous and feature very different pathophysiologies. Therefore, a thorough understanding of the underlying pathology as well as its specificity are critically needed to help optimize the management and anticipate complications⁴⁶. Moreover, patient selection and indication will need to be clearly defined, in order to select the patients who will benefit from this therapy the most. Here, we present a generalized mechanical strategy for diaphragm support, but the parameters of actuator design or actuation control will need to be optimized and specialized per the needs of a given pathology as well as individual patient anatomy.

Owing to the complexity of the procedure, a multidisciplinary team highly trained in advanced thoracic surgery is required to build expertise and develop this technology, ideally in a high-volume center⁴⁷. Technological improvement is required to provide the least invasive approach of implantation. In this regard, a thoracoscopic route might be beneficial and will be the subject of future work. Given the invasive nature of implantable devices, the diaphragm assist platform is targeted towards patients with chronic-to-permanent ventilator dependence. We recognize that surgery in patients suffering severe diaphragm dysfunction causing respiratory failure can carry a high morbidity and mortality. Peri-operative complications can be numerous; one of the most feared is the worsening of the pulmonary status, which may itself precipitate the need for long-term ventilation⁴⁸. Nevertheless, it has been well demonstrated that complex thoracic surgery is feasible even in very frail patients. Lung transplantation for terminal respiratory disease⁴⁹ is one of the most striking examples. Thus, surgery could still be considered in a suitable target population that would ultimately benefit from this mechanical augmentation of diaphragm function, such as a range of neuromuscular disorders. The concept of diaphragm assist is in itself a means of preventing further complications from chronic respiratory failure and preserving key aspects of quality of life-like speech and mobility.

Due to the focus on feasibility, we acknowledge that there are limitations in these acute studies from the lens of regulatory approval and clinical translation. We do not study device biocompatibility or long-term device operation. The device was constructed from types of polymers that are already used in established medical devices^{50–53}, such as poly(ethylene terephthalate) (PET) and polyurethanes (See Supplemental Information). Because our device focuses on mechanical interaction, as opposed to biochemical interactions with the body, the materials used in the device can easily be substituted with regulatory-approved materials in future iterations. With improved performance and stability, future long-term studies will need to investigate the long-term effects of the system including tissue remodeling and the ability to provide full-time respiratory support.

Although this technology requires further advancements in the net tidal volumes it can generate before it can fully match the ventilation capacity of a current mechanical ventilator, it is the first study to report the ability to rescue ventilation with an implantable ventilator. We envision further translational potential of this technology when combined with the development of smaller and more portable pneumatic energy sources^{54,55} as the field of soft robotics advances. With the integration of a portable pump and control system in the future, this technology could provide an additional level of patient autonomy via increased mobility. Motivated by the encouraging results

of this study, we believe this technology, with optimized design, has the potential to provide a radically different ventilation technology that preserves key metrics of quality of life for people with end-stage mechanical respiratory failure.

Rebuttal 2

As requested, we carefully considered comments of Reviewer 3. In the paragraph "Effect of synchronization on blood gas exchange", we clarified the number of subjects included for this analysis (comment 1.1). We agree that the acidemia observed in Fig. 5b is of mixed cause and we modified the text accordingly (comment 1.2). Although the discussion is quite long (comment 2), the current version has been drastically improved thanks the suggestions from both reviewers. We believe that shortening the discussion would affect its quality. Therefore, we didn't modify it, unless the Editor feels it is essential.